# Uni-directional Blending: Learning Robust Representations for Few-shot Action Recognition with Frame-level Ambiguities

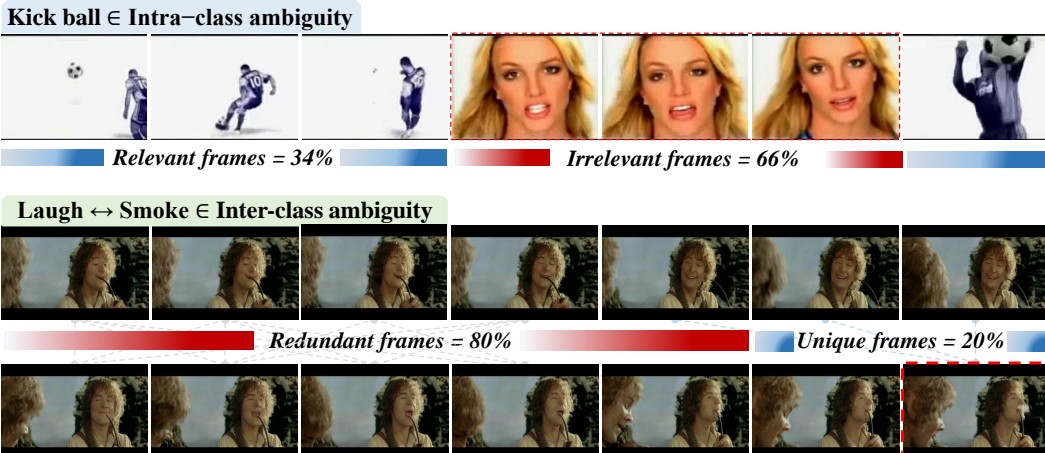

Figure 1: **Examples of frame-level ambiguities.** Intra-class ambiguity (e.g., *'kick ball'*) and inter-class ambiguity (e.g., *'laugh' and 'smoke'*).

## Abstract

Leveraging vision-language models (VLMs) for few-shot action recognition has shown promising results, yet direct image-text alignment methods, such as CLIP, encounter significant challenges in video domains due to frame-level ambiguities. Videos frequently include irrelevant and redundant frames, leading to intra-class ambiguity from non-essential content within the same action and inter-class ambiguity from visually overlapping elements across classes. These ambiguities hinder the learning of distinctive prototypes and robust semantic representations. To overcome this, we introduce Uni-FSAR, a novel framework that employs uni-directional blending to selectively integrate relevant frames, preventing contamination of prototypes by irrelevant visual noise. Additionally, a learnable text query (LTQ) bridges the semantic gap between visual features and class labels, enhancing representation alignment. Furthermore, our LTQ-based Semantic Bridging Loss promotes focus on informative frames through similarity-based gradient propagation, mitigating inter-class overlap and fostering more generalizable representations. Extensive experiments, including cross-dataset evaluations, demonstrate that Uni-FSAR achieves superior robustness in handling frame-level ambiguities compared to prior works. Quantitatively and qualitatively, our method outperforms the state-of-the-art by an average of 2.34% across benchmarks, with a notable 6.5% top-1 accuracy gain on HMDB51, where ambiguities are most pronounced.

## 1 Introduction

Understanding human actions in videos from diverse sources remains a core challenge in computer vision, as it requires reasoning over multi-frame context rather than relying on static visual cues from

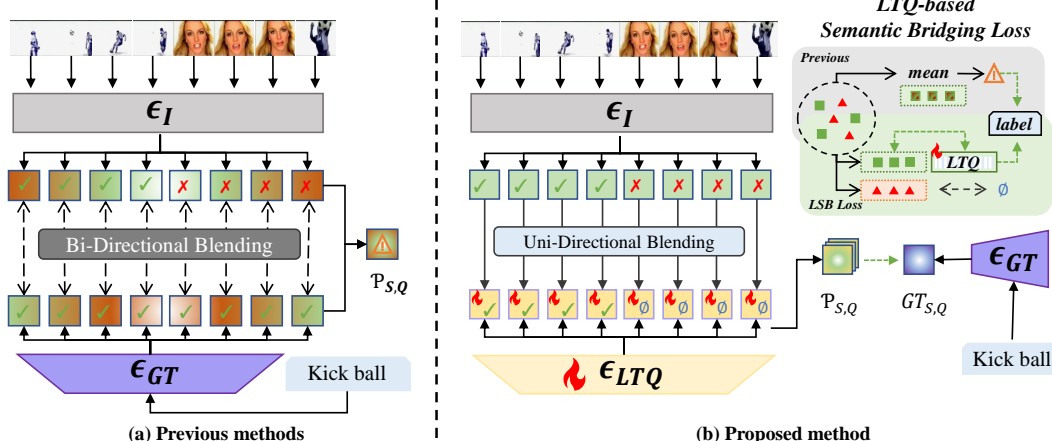

Figure 2: **Comparison of prototype alignment methods.** (a) Previous bi-directional methods mix informative (✓) and irrelevant (X) frames. (b) Our Uni-FSAR uses uni-directional blending. The **LSB Loss module (inset)** illustrates how we resolve the semantic gap: only informative features (green squares) are aligned with the LTQ, while irrelevant ones (red triangles) are filtered out with negligible gradients ($\emptyset$), preventing prototype contamination.

a single-frame. To address this, prior research has explored temporal modeling, such as capturing motion patterns or decomposing actions into finer sub-units (Cao et al., 2020; Feichtenhofer et al., 2019; Qu et al., 2024; Wang et al., 2023b; Yu et al., 2024; Zhao et al., 2022). While prior temporal modeling approaches (Cao et al., 2020; Feichtenhofer et al., 2019; Qu et al., 2024; Wang et al., 2023b; Yu et al., 2024; Zhao et al., 2022) have shown progress, they typically rely on large-scale closed-set datasets, struggling to generalize to novel classes (Dwivedi et al., 2019; Wang et al., 2023a). In contrast, few-shot action recognition (FSAR) addresses this by learning to recognize new classes from limited examples via meta-learning (Finn et al., 2017; Snell et al., 2017). Recently, incorporating vision-language models (VLMs) like CLIP (Radford et al., 2021) into FSAR has yielded notable gains by leveraging rich semantic alignment (Tang et al., 2024; Wang et al., 2023d; 2024). However, these methods often overlook *frame-level ambiguity* inherent in unconstrained videos (Carreira & Zisserman, 2017; Goyal et al., 2017; Kuehne et al., 2011; Soomro et al., 2012). Videos collected from unconstrained sources, such as YouTube, movies, frequently include irrelevant frames (e.g., background scenes, transitions) and redundant frames that appear across different action categories. These lead to two types of ambiguity: *intra-class ambiguity* due to irrelevant content within the same class, and *inter-class ambiguity* caused by overlapping content across classes. This issue is visually exemplified in Fig. 1, which shows how intra-class irrelevance and inter-class redundancy disrupt the semantic consistency of class representations. Rather than contributing to meaningful prototypes, these frames introduce visual noise and class overlap, making it difficult for the model to learn clear decision boundaries. Furthermore, we introduce the LTQ-based Semantic Bridging (LSB) Loss, which attenuates the influence of redundant frames through similarity-based gradient propagation.

Figure 2 illustrates a fundamental limitation in the prototype alignment process of prior FSAR methods. As shown in Fig. 2 (a), previous methods (Wang et al., 2024; Wu et al., 2024) adopt a bi-directional blending strategy where all frame-level features are indiscriminately aligned with class labels. This bi-directional blending incorporates both irrelevant and redundant frames into the prototype, thereby exacerbating intra- and inter-class ambiguities. To address these limitations, we propose a **Uni-FSAR** framework that aims to improve prototype construction under frame-level ambiguity by leveraging selective use of semantically relevant frame information. Figure 2 (b) depicts the core components of the proposed pipeline. We design a uni-directional blending strategy to prevent irrelevant frames from contaminating class prototypes. Unlike bi-directional blending where noisy frames can dominate the shared representation via attention feedback, our uni-directional design ensures that the Learnable Text Query (LTQ) aggregates visual evidence without broadcasting signals back into frame tokens. This effectively filters out visual noise while the LTQ semantically aligns visual features with class labels. Furthermore, we introduce the LTQ-based Semantic Bridging

(LSB) Loss, which attenuates the influence of redundant frames through similarity-based gradient propagation. The main contributions of this paper are summarized as follows:

- **Uni-directional blending strategy** and **Learnable Text Query (LTQ)** are designed to alleviate intra-class ambiguity caused by irrelevant frames, enabling more effective semantic alignment between visual representations and action labels.
- **LTQ-based Semantic Bridging (LSB) Loss** addresses inter-class ambiguity stemming from redundant frames by promoting selective focus on the most distinctive visual information through similarity-based gradient propagation.
- We propose **Uni-FSAR as a novel integrative framework** that synergistically combines uni-directional blending and LSB loss to enable effective multi-modal alignment, addressing frame-level ambiguities in diverse few-shot settings and achieving an average accuracy improvement of 2.34% with up to 6.5% gains across datasets, alongside strong cross-dataset generalizability.

## 2 RELATED WORK

**Human Action Recognition.** The action recognition field has progressed through innovations in network architectures such as 3D CNNs (Ji et al., 2012; Taylor et al., 2010; Tran et al., 2015) and transformers (Girdhar et al., 2019; Liu et al., 2022), as well as improvements in features incorporating additional modalities such as optical flow (Beauchemin & Barron, 1995; Horn & Schunck, 1981; Lee et al., 2018; Sevilla-Lara et al., 2019), skeleton data (Vemulapalli et al., 2014; Wang et al., 2013), and vision language models (Chen et al., 2023; Huang et al., 2024; Wang et al., 2021). Despite these advances, most approaches still operate under closed-set assumptions, where all classes are known during training. Such assumptions limit generalization to unseen actions, especially in real-world videos with temporal variation and frame-level ambiguity. To address this limitation, FSAR has been explored as an alternative to closed-set training, enabling generalization to novel actions from only a few labeled examples (Zhang et al., 2020).

**Few-shot Learning.** Few-shot learning has been primarily explored through tasks like image classification (Chowdhury et al., 2021; Snell et al., 2017), typically adopting an $N$-way $K$-shot setting, where $N$ is the number of classes, and $K$ is the number of labeled examples per class. This framework commonly adopts episodic training where models are trained on sampled support-query splits to improve generalization across tasks. In meta-learning, models learn a shared representation ($\phi$) across tasks and task-specific parameters or adaptation mechanisms ($\theta$), enabling fast adaptation and improved generalization to unseen tasks from limited examples. This approach has also been extended to few-shot action recognition in videos, where collecting large-scale annotations is particularly expensive.

**Vision-Language Models for FSAR.** Vision-language models, such as CLIP (Radford et al., 2021) and BLIP (Li et al., 2022), have recently been applied to FSAR to enhance generalization by leveraging semantically aligned image-text embeddings. These methods typically select frames that are most semantically similar to class labels in the embedding space and integrate text-tokenized class label features with frame-level visual features using attention mechanisms, mean pooling, or concatenation. These approaches have improved semantic alignment in FSAR and shown strong performance under limited supervision. However, they often overlook a core challenge in action recognition: frame-level ambiguity which can degrade prototype quality and lead to semantic overlap. Addressing this issue remains crucial for achieving robust generalization in real-world FSAR.

## 3 METHOD

### 3.1 PROBLEM DEFINITION

Few-shot action recognition aims to classify a query video into one of several previously unseen classes using only a small number of labeled examples per class. Unlike conventional action recognition, which relies on large-scale annotated data for all target classes, FSAR focuses on generalization to novel classes under limited supervision. This problem setting is particularly challenging in video data due to temporal complexity, large intra-class variation, and frame-level noise. The dataset is partitioned into disjoint subsets for training, validation, and testing, denoted as

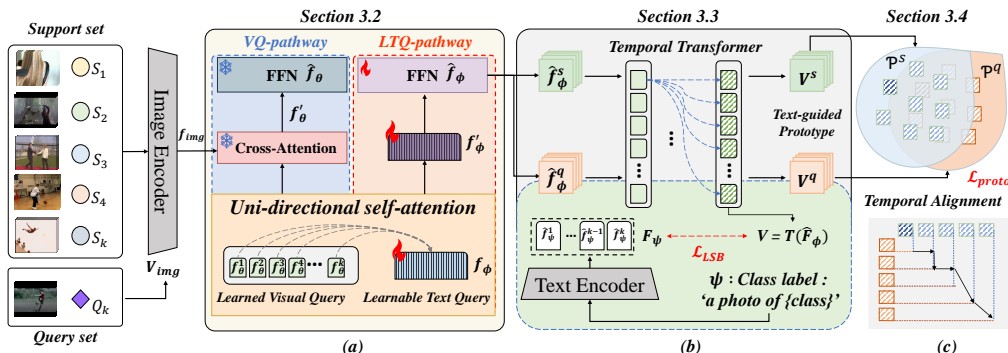

Figure 3: **Overview of our proposed Uni-FSAR**. Which consists of three main components: (a) UniQ-Former: uni-directional blending and LTQ generation, (b) temporal modeling and a semantic bridge module that connects visual features with class-level text representations, and (c) temporal alignment of support and query prototypes for few-shot classification.

$\mathcal{D}_{\text{train}}$, $\mathcal{D}_{\text{val}}$, and $\mathcal{D}_{\text{test}}$, respectively, where the corresponding class sets are mutually exclusive, i.e., $\mathcal{C}_{\text{train}} \cap \mathcal{C}_{\text{test}} = \emptyset$. To simulate the few-shot scenario, we adopt an episodic meta-learning framework. Each episode is constructed by sampling a support set $S_k = \{(x_i, y_i)\}_{i=1}^{N_k}$ for each class $k$, where $x_i$ denotes a video sample and $y_i$ its corresponding class label, along with a query set $Q$ from $\mathcal{D}_{\text{train}}$. This forms an $N$-shot $K$-way classification task. At test phase, episodes are constructed similarly using $\mathcal{D}_{\text{test}}$ or $\mathcal{D}_{\text{val}}$, with classes that were not observed during training. We follow a prototype-based classification strategy, where each class prototype $c_k$ is computed by averaging the support features and the query prediction is made by comparing the query feature to the nearest prototype as follows:

$$c_k = \frac{1}{N_k} \sum_{i=1}^{N_k} f(x_i) \qquad (1) \qquad \hat{y} = \arg\min_k d(f(x_q), c_k) \qquad (2)$$

where $f(\cdot)$ denotes the feature extractor and $d(\cdot, \cdot)$ is a similarity metric between the query and class prototype, computed using the temporal alignment module described in Sec. 3.4. We introduce the Uni-FSAR, illustrated in Fig. 3, which integrates uni-directional blending, the LTQ, and the LSB loss to address frame-level ambiguities effectively.

## 3.2 UNI-DIRECTIONAL BLENDING & LEARNABLE TEXT QUERY

To address the *intra-class ambiguity* challenge, frame-level interpretive capability is crucial for isolating relevant content in videos. Prior methods, such as bi-directional blending, align all video frames indiscriminately with the same text guide, leading to reduced frame-level discrimination, poorer generalization, and the recurring cost of generating manual text annotations for each prototype. To overcome these limitations, we propose a novel uni-directional blending scheme that enhances per-frame interpretation and learns temporal relations across frames, enabling efficient prototype generation without reliance on manual text guides. This is achieved by integrating a learnable text query (LTQ) that semantically bridges visual features and class labels, thereby focusing on informative content while mitigating contamination from irrelevant frames. To this end, we employ the Q-Former architecture to enable

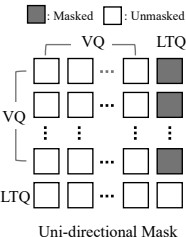

Figure 4: Uni-directional mask.

the LTQ to interpret frame-specific information. The LTQ $f_\phi \in \mathbb{R}^d$ is incorporated into the proposed UniQ-former as shown in Fig. 3 (a) to generate frame-wise text representation $\hat{f}_\phi \in \mathbb{R}^d$. This query is concatenated with the learned Visual Queries (VQ) $f_\theta \in \mathbb{R}^{32 \times d}$ of the UniQ-Former, which are trained using multiple loss functions to capture diverse image information. The concatenated queries $f_{\text{concat}} \in \mathbb{R}^{33 \times d}$ are fed into a self-attention module, where a uni-directional attention mask $M_{\text{uni}} \in \mathbb{R}^{33 \times 33}$ (Fig. 4), is applied to prevent the LTQ's information from being incorporated into the VQ. Specifically, $M_{\text{uni}}$ is a block mask that allows queries (including LTQ) to attend to VQ keys but masks all attention to the LTQ key, ensuring uni-directional flow from visual to text queries. By

blocking the attention feedback from LTQ to VQs, this mechanism prevents class semantics from overwriting frame features. Consequently, irrelevant frames are not forced to align with the text, effectively preventing them from contaminating the shared prototype.

$$f'_{\text{concat}} = \text{MHSA}(f_{\text{concat}}; M_{\text{uni}}) \tag{3}$$

where MHSA denotes the multi-head self-attention module, and the output $f'_{\text{concat}} \in \mathbb{R}^{33 \times d}$ is separated into $f'_\theta \in \mathbb{R}^{32 \times d}$ and $f'_\phi \in \mathbb{R}^d$. The VQs ($f'_\theta$) attend to the image features $f_{\text{img}}$ through a cross-attention module, and the resulting output is processed by the feed-forward network (FFN) in the *VQ-pathway* to produce semantically rich and diverse VQs ($\hat{f}_\theta$) $\in \mathbb{R}^{32 \times d}$.

$$\hat{f}_\theta = \text{FFN}_{\text{VQ}}(\text{MHCA}(f'_\theta, f_{\text{img}})) \tag{4}$$

where MHCA represents multi-head cross attention module. After incorporating rich information via Eq. 3, $f'_\phi$ is processed by the FFN of the *LTQ-pathway*, yielding $\hat{f}_\phi \in \mathbb{R}^d$, which capture comprehensive frame representation as follows:

$$\hat{f}_\phi = \text{FFN}_{\text{LTQ}}(f'_\phi) \quad (5) \qquad V = \mathcal{T}(\hat{F}_\phi) \quad (6)$$

The features of a video sample, $\hat{F}_\phi = [\hat{f}_\phi^{(1)}, \hat{f}_\phi^{(2)}, ..., \hat{f}_\phi^{(t)}]$, are computed using Eq 5. These features represent individual frames and to capture temporal relationships between frames, a multi-layer transformer $\mathcal{T}$ is employed. This transformer explicitly models cross-frame dependencies, allowing the overall pipeline to capture motion cues beyond static appearances. This produces video features $V \in \mathbb{R}^{t \times d}$, where $t$ denotes the number of frames and $d$ represents the feature dimension.

### 3.3 LTQ-BASED SEMANTIC BRIDGING LOSS

To mitigate the *inter-class ambiguity* inherent in unconstrained videos, where noisy and redundant frames often coexist with annotated actions, indiscriminately aligning all frames to a single label exacerbates class overlap and weakens prototype distinctiveness. To overcome this, we propose the LTQ-based Semantic Bridging Loss (LSB Loss), which employs a contrastive formulation to selectively align only the K frames most semantically similar to the class label. We use UniQ-Former to extract the text feature of label as follow:

$$f'_\psi = \text{FFN}_{\text{LTQ}}(\text{MHSA}(f_\psi)), \quad \text{where} \quad f_\psi = \text{Embedding}(\psi) \tag{7}$$

where the $f_\psi \in \mathbb{R}^{s \times d}$ represents the embeddings of text prompt $\psi$, and $f'_\psi \in \mathbb{R}^{s \times d}$ denotes the resulting text feature. When the set of all [CLS] tokens of text features is denoted as $F_\psi \in \mathbb{R}^{N \times d}$, the cosine similarity is computed between video features $V$ and text features $F_\psi$. For each class, the K frames with the highest similarity are selected. We set $K=3$ based on ablation studies (see Table 6). Frames not selected by Top-$K$ do not contribute to the LSB objective and receive zero gradient, and their average is computed as follows:

$$A = \text{Similarity}(F_\psi, V) \quad (8) \qquad A' = \text{Top-K}(A) \quad (9) \qquad f_\delta = \text{Mean}(A') \quad (10)$$

By applying the softmax to $f_\delta$, the probability distribution $\hat{p}_{\text{LSB}}$ for the target text features is obtained, and the cross-entropy loss is computed as follows:

$$\mathcal{L}_{\text{LSB}} = -\sum_{i=1}^{N} p^{(i)} \log \hat{p}_{\text{LSB}}^{(i)}, \quad \text{where} \quad \hat{p}_{\text{LSB}}^{(i)} = \frac{\exp(f_\delta^{(i)}/\tau)}{\sum_{j=1}^{N} \exp(f_\delta^{(j)}/\tau)} \tag{11}$$

where $p$ denotes the ground-truth probability distribution, $\tau$ is a learnable temperature parameter, and the $N$ denotes the number of action classes. By training the model with the LSB loss, it can align selected frames with the video's text prompt, thereby contributing to mitigating the inter-class ambiguity problem.

### 3.4 PROTOTYPE METRIC-BASED ALIGNMENT

Given an $N$-shot $K$-way support set $S_k = \{(x_i, y_i)\}_{i=1}^{N_k}$, samples from the support set are processed by the model and the output, as defined in Eq. 6, constitutes the features of support set $\hat{S}_k = \{V_i\}_{i=1}^{N_k}$

for the $k$-th class. The $k$-th prototype $\mathcal{P}_k$ is is generated by computing the mean of all elements in $\hat{S}_k$, representing the characteristic feature of the $k$-th class. Similarly, video features $V_q \in \mathbb{R}^{t \times d}$ for a query sample are extracted using Eq. 6. In prototype learning, to classify a query sample, the distances between its features $V_q$ and each prototype $\mathcal{P}_k$ are calculated. To account for the temporal order of video frames in distance computation, we employ the OTAM (Cao et al., 2020).

$$\mathcal{P}_k = \frac{1}{N_k} \sum_{i=1}^{N_k} V_i^s \quad (12) \qquad d_k = \text{OTAM}(V^q, \mathcal{P}_k) \quad (13)$$

where the $\mathcal{P}_k \in \mathbb{R}^{t \times d}$ denotes the prototype of the $k$-th class, $V_i^s$ and $V^q$ represent the video features of the $i$-th support sample and the query sample, respectively, computed using Eq. 6, $N_k$ denotes the number of samples for the $k$-th class, and the $d_k$ represents the distance between the query features and the $k$-th prototype. Crucially, OTAM operates on the **full sequence** ($V^q, \mathcal{P}_k$), utilizing *all* frames. LSB acts solely as a gradient modulator on Top-$K$ frames, ensuring that temporal dynamics remain intact without information loss. After calculating distance between a query and the prototypes, the probability distribution $\hat{p}_{\text{proto}}$ of a query belonging to each class is derived based on these distances. The prototype metric-based alignment loss is computed as follows:

$$\mathcal{L}_{\text{proto}} = -\sum_{i=1}^{K} p_i \log \hat{p}_{\text{proto}}(d_i), \quad \text{where} \quad \hat{p}_{\text{proto}}(d_k) = \frac{\exp(-d_k)}{\sum_{i=1}^{N_k} \exp(-d_i)} \quad (14)$$

where $\hat{p}_{\text{proto}}(d_k)$ represents the probability that the query belongs to the $k$-th class. According to Eq. 14, the query is trained to align with the prototype of its true class. The overall loss, combining Eq. 11 and Eq. 14, is computed as follows:

$$\mathcal{L} = \mathcal{L}_{\text{LSB}} + \alpha \mathcal{L}_{\text{proto}} \quad (15)$$

where $\alpha$ is a weighting factor that balances the contributions of the two loss components. We provide a sensitivity analysis in Appendix A.5, demonstrating that the model performance remains stable and robust to variations in $\alpha$ By training with Eq. 15, the meta-parameter $f_\phi$ is optimized to extract video representations robust to intra- and inter-class ambiguities. Additional explanations of the algorithms are provided in Appendix A.1.

## 4 EXPERIMENTS

### 4.1 DATASETS AND EXPERIMENT SETUPS

**Datasets.** We conducted experiments on five datasets: UCF101 (Soomro et al., 2012), Kinetics100 (Carreira & Zisserman, 2017), HMDB51(Kuehne et al., 2011), Something-Something V2 small (Goyal et al., 2017; Zhu & Yang, 2018), to evaluate performance fairly. UCF101, Kinetics100, and HMDB51 feature third person views of daily actions such as walking and sports sourced from public media. Conversely, Something-Something V2 (SSv2) captures egocentric object interactions. We therefore prioritize benchmarks where frame-level ambiguity is intrinsic (UCF101, Kinetics100, HMDB51), and include SSv2 solely for fairness, enabling a faithful evaluation of our problem formulation. Details are provided in Appendix A.2.

**Implementation Details.** The proposed model employs a pre-trained ViT-L/14 as the image encoder and a Q-Former (Li et al., 2023). For training, the Adam optimizer is employed with a single warm-up epoch. During inference, the model was evaluated by computing the average accuracy over 10,000 randomly sampled episodes. Details of these hyperparameters are provided in the Appendix A.2.

### 4.2 COMPARISON WITH STATE-OF-THE-ART METHODS

To verify the effectiveness of the proposed framework, we compare the performance of our Uni-FSAR model with current state-of-the-art few-shot action recognition methods across five standard benchmarks under 5-way $K$-shot setting. The results are summarized in Tab. 1 and Tab. 2. In particular, we present a fair and quantitative evaluation under multi-modal settings by comparing

Table 1: Comparison with state-of-the-art methods on the UCF101, Kinetics.

| Method | Reference | Backbone | UCF101 | | | Kinetics | | |
|---|---|---|---|---|---|---|---|---|
| | | | 1-shot | 3-shot | 5-shot | 1-shot | 3-shot | 5-shot |
| OTAM (Cao et al., 2020) | CVPR'20 | INet-RN50 | 79.9 | 87.0 | 88.9 | 72.2 | 78.7 | 84.2 |
| TRX (Perrett et al., 2021) | CVPR'21 | INet-RN50 | 78.2 | 92.4 | 96.1 | 63.6 | 80.1 | 85.2 |
| STRM (Thatipelli et al., 2022) | CVPR'22 | INet-RN50 | 80.5 | 92.7 | 96.9 | 62.9 | - | 86.7 |
| HyRSM (Wang et al., 2022) | CVPR'22 | INet-RN50 | 83.9 | 93.0 | 94.7 | 73.7 | - | 86.1 |
| HCL (Zheng et al., 2022) | ECCV'22 | INet-RN50 | 82.5 | 91.0 | 93.9 | 73.7 | - | 85.8 |
| MoLo (OTAM) (Wang et al., 2023c) | CVPR'23 | INet-RN50 | 85.4 | 93.4 | 95.1 | 73.8 | - | 85.1 |
| OTAM$^\dagger$ (Cao et al., 2020) | CVPR'20 | BLIP$_{ViT-B}$ | 91.4 | - | 96.5 | 82.4 | - | 91.1 |
| TRX$^\dagger$ (Perrett et al., 2021) | CVPR'21 | BLIP$_{ViT-B}$ | 90.9 | - | 97.4 | 76.6 | - | 90.8 |
| HyRSM$^\dagger$ (Wang et al., 2022) | CVPR'22 | BLIP$_{ViT-B}$ | 91.6 | - | 96.9 | 82.4 | - | 91.8 |
| BLIP-Freeze$_{visual}$ (Li et al., 2022) | ICML'22 | BLIP$_{ViT-B}$ | 88.9 | - | 95.3 | 74.8 | - | 87.5 |
| BLIP-Freeze$_{text}$ (Li et al., 2022) | ICML'22 | BLIP$_{ViT-B}$ | 86.4 | - | 95.1 | 72.9 | - | 86.5 |
| CapFSAR (OTAM) (Wang et al., 2023d) | arXiv'23 | BLIP$_{ViT-B}$ | 93.3 | - | 97.8 | 84.9 | - | 93.1 |
| EMP-Net (Wu et al., 2024) | ECCV'24 | CLIP$_{ViT-B}$ | 94.3 | - | 98.2 | - | - | - |
| CLIP-FSAR (Wang et al., 2024) | IJCV'24 | CLIP$_{ViT-B}$ | 96.6 | 98.4 | 99.0 | 89.7 | 94.2 | 95.0 |
| **Ours (Uni-FSAR)** | - | BLIPv2$_{ViT-L}$ | **97.5** | **98.8** | **99.0** | **92.8** | **95.7** | **96.6** |

Table 2: Comparison with state-of-the-art methods on the SSv2-Small and HMDB51.

| Method | Reference | Backbone | SSv2-Small | | | HMDB51 | | |
|---|---|---|---|---|---|---|---|---|
| | | | 1-shot | 3-shot | 5-shot | 1-shot | 3-shot | 5-shot |
| OTAM (Cao et al., 2020) | CVPR'20 | INet-RN50 | 36.4 | 45.9 | 48.0 | 54.5 | 65.7 | 68.0 |
| TRX (Perrett et al., 2021) | CVPR'21 | INet-RN50 | 36.0 | 51.9 | 56.7 | 53.1 | 66.8 | 75.6 |
| STRM (Thatipelli et al., 2022) | CVPR'22 | INet-RN50 | 37.1 | 49.2 | 55.3 | 52.3 | 67.4 | 77.3 |
| HyRSM (Wang et al., 2022) | CVPR'22 | INet-RN50 | 40.6 | 52.3 | 56.1 | 60.3 | 71.7 | 76.0 |
| HCL (Zheng et al., 2022) | ECCV'22 | INet-RN50 | 38.7 | 49.1 | 55.4 | 59.1 | 71.2 | 76.3 |
| MoLo (OTAM) (Wang et al., 2023c) | CVPR'23 | INet-RN50 | 41.9 | 50.9 | 56.2 | 59.8 | 71.1 | 76.1 |
| OTAM$^\dagger$ (Cao et al., 2020) | CVPR'20 | BLIP$_{ViT-B}$ | 45.5 | - | | 63.9 | - | 76.5 |
| TRX$^\dagger$ (Perrett et al., 2021) | CVPR'21 | BLIP$_{ViT-B}$ | 40.6 | - | 61.0 | 58.9 | - | 79.9 |
| HyRSM$^\dagger$ (Wang et al., 2022) | CVPR'22 | BLIP$_{ViT-B}$ | 45.5 | - | 60.7 | 69.8 | - | 80.6 |
| BLIP-Freeze$_{visual}$ (Li et al., 2022) | ICML'22 | BLIP$_{ViT-B}$ | 31.2 | - | 40.3 | 56.2 | - | 72.8 |
| BLIP-Freeze$_{text}$ (Li et al., 2022) | ICML'22 | BLIP$_{ViT-B}$ | 28.7 | - | 39.5 | 52.4 | - | 67.2 |
| CapFSAR (OTAM) (Wang et al., 2023d) | arXiv'23 | BLIP$_{ViT-B}$ | 45.9 | - | 59.9 | 65.2 | - | 78.6 |
| EMP-Net (Wu et al., 2024) | ECCV'24 | CLIP$_{ViT-B}$ | **57.1** | - | 65.7 | 76.8 | - | 85.8 |
| CLIP-FSAR (Wang et al., 2024) | IJCV'24 | CLIP$_{ViT-B}$ | 54.5 | 58.6 | 61.8 | 75.8 | 84.1 | 87.7 |
| **Ours (Uni-FSAR)** | - | BLIPv2$_{ViT-L}$ | 54.1 | **64.4** | **68.8** | **82.3** | **88.4** | **90.5** |

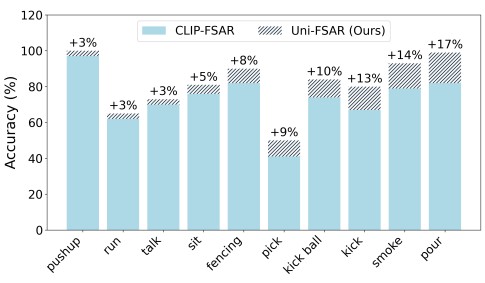

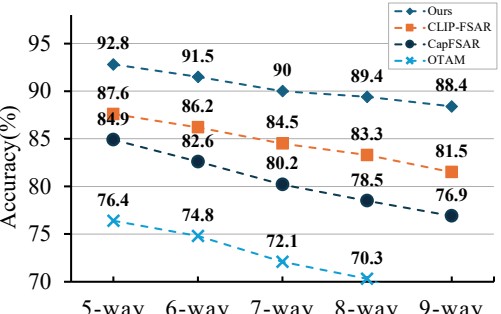

Figure 5: Comparison of quantitative results on class-wise performance of HMDB51.

Figure 6: Comparison of performance under the $N$-way 1-shot setting on the Kinetics dataset.

with recent models that utilize VLMs such as CLIP and BLIP (Radford et al., 2021; Li et al., 2022). Based on these results, we present two key observations:

As shown in Tab. 1 and Tab. 2, our model achieves state-of-the-art performance on UCF101, Kinetics, HMDB51 and SSv2-small datasets. In particular on HMDB51, our model achieves significant improvements of +6.5% in the 1-shot setting, +4.3% in 3-shot, and +2.8% in 5-shot compared to the previous best method. These improvements highlight the robustness of the proposed Uni-FSAR, especially on datasets with a noisy video samples. And supports our hypothesis that intra- and inter class ambiguities in dataset are critical challenge and demonstrates the effectiveness of our uni-directional blending approach using LTQ and the LSB Loss.

For the SSv2-small dataset, our model achieves the highest performance in the 3 and 5-shot settings, with a comparable performance in the 1-shot case for the SSv2-small. To further investigate this,

Table 3: Cross-dataset validation results.

| Source → Target | Method | 1-shot / 5-shot |
|---|---|---|
| HMDB51 → Kinetics | CLIP-FSAR | 75.5 / 86.7 |
| | Ours | **88.6 / 94.9** |
| HMDB51 → SSv2-small | CLIP-FSAR | 33.9 / 46.5 |
| | Ours | **52.3 / 68.2** |
| SSv2-small → HMDB51 | CLIP-FSAR | 37.1 / 46.3 |
| | Ours | **72.7 / 85.0** |

Table 4: Ablation of each module on HMDB51 and SSv2-Small.

| Uni-dir. blend & LTQ | LTQ-based Bridging Loss | HMDB51 1-shot | HMDB51 5-shot | SSv2-Small 1-shot | SSv2-Small 5-shot |
|---|---|---|---|---|---|
| – | – | 67.0 | 81.5 | 40.5 | 54.3 |
| – | ✓ | 67.0 | 81.6 | 41.3 | 55.8 |
| ✓ | – | 80.2 | 89.9 | 52.2 | 68.2 |
| ✓ | ✓ | **82.3** | **90.5** | **54.1** | **68.8** |

Table 5: Ablation study on the effect of different numbers of Learnable Text Queries.

| LTQ ($T_Q$) | HMDB51 1-shot | HMDB51 5-shot | SSv2-Small 1-shot | SSv2-Small 5-shot |
|---|---|---|---|---|
| $T_Q = 1$ (Default) | 82.3 | 90.5 | **54.1** | **68.8** |
| $T_Q = 4$ | **82.5** | **90.8** | 51.3 | 67.8 |
| $T_Q = 8$ | 81.8 | 90.1 | 52.4 | 67.3 |

Table 6: Ablation study on different semantic bridging strategies.

| Semantic Bridging Strategy | HMDB51 1-shot | HMDB51 5-shot | SSv2-Small 1-shot | SSv2-Small 5-shot |
|---|---|---|---|---|
| GAP + Mean | 81.6 | 90.1 | 52.4 | 68.3 |
| LSB (Top-1) | 81.9 | 90.2 | 53.5 | 68.1 |
| LSB (Top-3)(Default) | **82.3** | **90.5** | **54.1** | **68.8** |

Table 7: Ablation study on different type of prompt.

| Prompt types | HMDB51 1-shot | HMDB51 5-shot | SSv2-Small 1-shot | SSv2-Small 5-shot |
|---|---|---|---|---|
| {} (None) | 81.7 | 90.3 | 51.9 | 67.3 |
| 'a photo of' (Default) | **82.3** | **90.5** | **54.1** | **68.8** |
| Learnable | 81.4 | 90.4 | 50.7 | 67.2 |

Table 8: Ablation study on different number of frames.

| # of frames | HMDB51 1-shot | HMDB51 5-shot | SSv2-Small 1-shot | SSv2-Small 5-shot |
|---|---|---|---|---|
| 4 | 80.0 | 89.1 | 49.4 | 63.3 |
| 8 | **82.3** | **90.5** | **54.1** | **68.8** |
| 12 | 82.1 | 90.5 | 52.8 | 68.5 |

we conducted a cross-dataset generalization experiment with CLIP-FSAR. As shown in Tab. 3, although our method shows slightly lower performance than CLIP-FSAR in within-dataset training on SSv2, it achieves notably better results in the cross-dataset validation setting. In particular, our method outperforms CLIP-FSAR by a large margin in challenging scenarios such as transferring from HMDB51 to SSv2-small, achieving a +18.4% gain in the 1-shot setting. This suggests that CLIP-FSAR is more prone to overfitting to single-dataset distributions, whereas our method demonstrates stronger robustness and generalizability across domains. Additional quantitative analyses are provided in the Appendix A.5. In particular, we address the backbone fairness concern and demonstrate in a backbone-controlled setting that most of the performance gains stem from our Uni-FSAR modules rather than backbone scaling.

## 4.3 ABLATION STUDY

We conduct a systematic analysis of the contributions of each module in our method. Table 4 presents the ablation results assessing the individual and combined effects of Uni-directional blending & LTQ and LTQ-based Semantic Bridging Loss. Applying either module alone yields performance improvements over the baseline, with Uni-directional blending & LTQ contributing more significantly—especially on the target dataset HMDB51 (+15.3% in 1-shot), and even on the more challenging SSv2-Small (+13.6% in 1-shot). When both modules are combined, the best performance is achieved across all settings, demonstrating their complementary benefits in enhancing FSAR.

Figure 6 presents accuracy trends for various few-shot action recognition methods as the number of classes (N-way) increases from 5 to 9. Our method consistently outperforms all baselines, maintaining the highest accuracy across all settings. Previous baselines exhibit significant drops as N increases, indicating limited scalability.

**The Impact of Uni-Directional Blending & LTQ.** As shown in Fig. 5, the proposed method achieves notable improvements exceeding +10% gain in classes such as 'pick', 'kick ball', 'smoke', and 'pour', where frame-level ambiguities are frequent and require fine-grained contextual understanding. These results validate the robustness of our selective prototype construction. In addition, Tab. 5 shows how the number of LTQs impacts performance. The results demonstrate that while the optimal setting may vary across cases, the default configuration generally yields the best overall performance.

**The Impact of LTQ-based Semantic Bridging Loss.** To evaluate the effectiveness of different prototype selection strategies, we compare the performance of GAP and LSB losses with Top-1

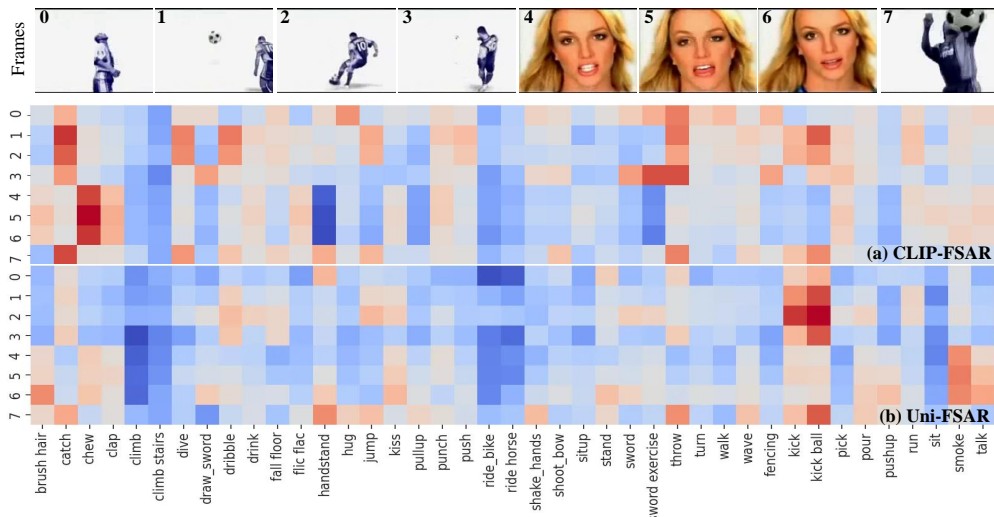

Figure 7: Comparison of qualitative result on noisy frame included sample of HMDB51.

and 3 setting under both 1-shot and 5-shot settings, as shown in Tab. 6. Overall, the LSB with Top-3 methods outperform GAP+Mean method Wang et al. (2024) in all scenarios, indicating that leveraging frame-level ranking information contributes to more discriminative prototype construction. We set the Top-K=3 once based on Tab. 6 and use the same value across all datasets and cross-dataset evaluations.

**The Impact of Prompt design and Number of Frames.** As shown in Tab. 7, we conduct the experiment on different types of prompt types to verify its domain inherent gap. As the tables shows, the default setting shows best performance between different settings. Table 8 shows, the different number of frames setting when using our model, typically more frames leading to better temporal information understanding, but for the CLIP-FSAR the performance is rather shows degradation.

## 4.4 Qualitative Results

We visualize the model's frame-wise predictions on video samples from HMDB51 to demonstrate the contextual recognition capability of our framework. As shown in Fig. 7, we compare class activation patterns between CLIP-FSAR and our proposed Uni-FSAR on a sample containing noisy frames. As shown in Fig. 7 (a), CLIP-FSAR exhibits noisy and inconsistent activations, with high confidence in false positives such as 'chew' (a training class) on irrelevant face frames, and misclassifies other frames as 'throw' or 'catch' despite the target action being 'kick ball'. In contrast, Uni-FSAR (Fig. 7 (b)) consistently activates only on the correct class, 'kick ball', while effectively ignoring distractors, indicating superior generalization to unseen classes and robustness to irrelevant visual content. Additional qualitative analyses across various samples are provided in the Appendix A.6.

## 5 Limitation

In this work, we focused on bridging context between visual inputs and action labels to address ambiguities at the frame level. Considering the inherent limitations of vision-language models (VLMs), such as their limited frame input capacity (e.g., approximately 12 frames) and the need for lightweight model deployment, future extensions should explore more efficient architectures. This increased frame capacity would subsequently enable more detailed modeling of temporal dynamics and spatial reasoning within individual frames, thereby tackling the limitations observed on SSv2 and enhancing overall spatio-temporal relationship understanding.

## 6 CONCLUSION

We introduced Uni-FSAR, a novel framework for few-shot action recognition that addresses frame-level ambiguities by combining uni-directional blending, the learnable text query, and the semantic bridging loss. Our method selectively aligns informative frames to improve prototype construction, effectively mitigating both intra-class and inter-class ambiguities. Extensive experiments across multiple benchmarks validate the robustness and generalizability of our approach, achieving state-of-the-art performance under both within-domain and cross-dataset settings. We believe this work provides a strong foundation for future research on semantically grounded prototype learning and label-aware visual reasoning in real-world video understanding scenarios. In particular, our findings highlight the importance of bridging visual-language semantics at the frame level, beyond conventional feature aggregation. Future work may extend this direction by incorporating fine-grained spatio-temporal modeling and scalable lightweight architectures for broader applicability.

## 7 REPRODUCIBILITY STATEMENT

To ensure reproducibility, we provide detailed descriptions of our methods in Section 3, including the mathematical formulations and algorithms outlined in **Appendix A.1**. Detailed information on the hyperparameter settings, dataset processing, and preprocessing steps used in the experiments is provided in **Appendix A.2**. The computational resources employed for all experiments are described in **Appendix A.4**. To address performance variance due to randomness, we report results averaged over multiple independent runs with fixed random seeds (detailed in **Appendix A.3**), and experiments on statistical significance are presented in Table 11. For full reproducibility, the source code will be made publicly available upon acceptance.

## 8 ETHICS STATEMENT

All authors have read and adhere to the ICLR Code of Ethics. Our study uses only public benchmarks (UCF101, HMDB51, Kinetics-100 from Kinetics-400, SSv2-Small); no new data were collected, no human subjects were recruited, and no personally identifiable information beyond public releases was used (no IRB needed). We follow dataset licenses and do not redistribute data. We caution against deployment in privacy-sensitive settings without lawful basis and risk assessment. Acknowledging possible dataset biases, we report results across multiple datasets and will release code/configs for reproducibility and independent auditing. We report compute/hyperparameters to support energy estimation and favor efficient settings when possible. The authors declare no conflicts of interest.

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

# A  APPENDIX

**Overview.** The supplementary includes the following sections:

## A.1  ALGORITHMS FOR TRAINING AND TESTING

---

**Algorithm 1** Training

---

1: **Input:** Training set $\mathcal{D}_{\text{train}}$, image encoder $\mathcal{V}_{\text{img}}(\cdot)$, UniQ-Former $\mathcal{Q}(\cdot)$, temporal transformer $\mathcal{T}(\cdot)$, VQ $f_\theta \in \mathbb{R}^{32 \times d}$, LTQ $f_\phi \in \mathbb{R}^d$, uni-directional attention mask $M_{\text{uni}} \in \mathbb{R}^{32 \times 32}$, ground-truth text prompts $\Psi = \{\Psi_c\}_{c=1}^C$, temperature $\tau$, loss weight $\alpha$, learning rate $\eta$

2:    *# Extract text features using Q-Former*

3:    $f_\Psi \leftarrow \text{Embedding}(\Psi)$

4:    $f'_\Psi \leftarrow \mathcal{Q}(f_\Psi)$               $\triangleright f'_\Psi \in \mathbb{R}^{C \times d}$

5: **for all** episodes in $\mathcal{D}_{\text{train}}$ **do**

6:    **for all** support set $S = \{X^s, Y^s\}$, query set $Q = \{X^q, Y^q\}$ and text features $f'_\psi \in \mathbb{R}^{K \times d}$ in an episode **do**

7:       *# Extract features using image encoder*

8:       $F^s_{\text{img}} \leftarrow \mathcal{V}_{\text{img}}(X^s)$     $\triangleright F^s_{\text{img}} \in \mathbb{R}^{(N \times K) \times t \times n \times d}$

9:       $F^q_{\text{img}} \leftarrow \mathcal{V}_{\text{img}}(X^q)$     $\triangleright F^q_{\text{img}} \in \mathbb{R}^{N_q \times t \times n \times d}$

10:     *# Extract features using Q-Former*

11:     $\hat{F}^s_\phi \leftarrow \mathcal{Q}(F^s_{\text{img}}; f_\theta, f_\phi, M_{\text{uni}})$     $\triangleright \hat{F}^s_\phi \in \mathbb{R}^{(N \times K) \times t \times d}$

12:     $\hat{F}^q_\phi \leftarrow \mathcal{Q}(F^q_{\text{img}}; f_\theta, f_\phi, M_{\text{uni}})$     $\triangleright \hat{F}^q_\phi \in \mathbb{R}^{N_q \times t \times d}$

13:     *# Process temporal transformer*

14:     $V^s, V^q \leftarrow \mathcal{T}(\hat{F}^s_\phi), \mathcal{T}(\hat{F}^q_\phi)$     $\triangleright V^s \in \mathbb{R}^{(N \times K) \times t \times d}, V^q \in \mathbb{R}^{N_q \times t \times d}$

15:     $V \leftarrow [V^s, V^q]$     $\triangleright V \in \mathbb{R}^{((N \times K) + N_q) \times t \times d}$

16:     *# Compute text features and select top-$\kappa$ frames*

17:     $A \leftarrow \text{CosineSimilarity}(V, f'_\psi)$     $\triangleright A \in \mathbb{R}^{((N \times K) + N_q) \times t \times K}$

18:     $A' \leftarrow \text{Top-}\kappa(S)$     $\triangleright A' \in \mathbb{R}^{((N \times K) + N_q) \times \kappa \times K}$

19:     $f_\delta \leftarrow \text{Mean}_\kappa(\mathcal{T})$     $\triangleright f_\delta \in \mathbb{R}^{((N \times K) + N_q) \times K}$

20:     $\hat{p}_{\text{LSB}} \leftarrow \frac{\exp(f_\delta / \tau)}{\sum_{k=1}^K \exp(f_{\delta, k} / \tau)}$     $\triangleright \hat{p}_{\text{LSB}} \in \mathbb{R}^{((N \times K) + N_q) \times K}$

21:     *# Compute LSB loss*

22:     $Y \leftarrow [Y^s, Y^q]$     $\triangleright Y \in \mathbb{R}^{((N \times K) + N_q)}$

23:     $\mathcal{L}_{\text{LSB}} \leftarrow \text{CrossEntropy}(Y, \hat{p}_{\text{LSB}})$

24:     *# Generate prototypes and compute distances*

25:     $\mathcal{P}_k \leftarrow \frac{1}{N} \sum_{i=1}^N V^s_{i,k}, \quad \forall k \in \{1, \ldots, K\}$     $\triangleright \mathcal{P}_k \in \mathbb{R}^{K \times t \times d}$

26:     $D_k \leftarrow \text{OTAM}(V^q, \mathcal{P}_k) \quad \forall k \in \{1, \ldots, K\}$     $\triangleright D \in \mathbb{R}^{N_q \times K}$

27:     *# Compute prototype-based probability and loss*

28:     $\hat{p}_{\text{proto}}(D_k) \leftarrow \frac{\exp(-D_k)}{\sum_{i=1}^K \exp(-D_i)}, \quad \forall k \in \{1, \ldots, K\}$     $\triangleright \hat{p}_{\text{proto}} \in \mathbb{R}^{N_q \times K}$

29:     $\mathcal{L}_{\text{proto}} \leftarrow \text{CrossEntropy}(Y^q, \hat{p}_{\text{proto}})$

30:     *# Compute overall loss and update parameters*

31:     $\mathcal{L} \leftarrow \mathcal{L}_{\text{LSB}} + \alpha \mathcal{L}_{\text{proto}}$

32:     $w_{t+1} \leftarrow w_t - \eta \nabla_w \mathcal{L}$

33:    **end for**

34: **end for**

---

---

**Algorithm 2** Testing

---

1: **Input:** Test set $\mathcal{D}_{\text{test}}$, image encoder $\mathcal{V}_{\text{img}}(\cdot)$, UniQ-Former $\mathcal{Q}(\cdot)$, temporal transformer $\mathcal{T}(\cdot)$, VQ $f_\theta \in \mathbb{R}^{32 \times d}$, LTQ $f_\phi \in \mathbb{R}^d$, uni-directional attention mask $M_{\text{uni}} \in \mathbb{R}^{32 \times 32}$, temperature $\tau$

2: **for all** episodes in $\mathcal{D}_{\text{test}}$ **do**

3:     **for all** support set $S = \{X^s, Y^s\}$ and query set $Q = \{X^q, Y^q\}$ in an episode **do**

4:         *# Extract features using image encoder*

5:         $F_{\text{img}}^s \leftarrow \mathcal{V}_{\text{img}}(X^s)$                   $\triangleright F_{\text{img}}^s \in \mathbb{R}^{(N \times K) \times t \times n \times d}$

6:         $F_{\text{img}}^q \leftarrow \mathcal{V}_{\text{img}}(X^q)$                   $\triangleright F_{\text{img}}^q \in \mathbb{R}^{N_q \times t \times n \times d}$

7:         *# Extract features using Q-Former*

8:         $\hat{F}_\phi^s \leftarrow \mathcal{Q}(F_{\text{img}}^s; f_\theta, f_\phi, M_{\text{uni}})$       $\triangleright \hat{F}_\phi^s \in \mathbb{R}^{(N \times K) \times t \times d}$

9:         $\hat{F}_\phi^q \leftarrow \mathcal{Q}(F_{\text{img}}^q; f_\theta, f_\phi, M_{\text{uni}})$       $\triangleright \hat{F}_\phi^q \in \mathbb{R}^{N_q \times t \times d}$

10:       *# Process temporal transformer*

11:       $V^s, V^q \leftarrow \mathcal{T}(\hat{F}_\phi^s), \mathcal{T}(\hat{F}_\phi^q)$     $\triangleright V^s \in \mathbb{R}^{(N \times K) \times t \times d}, V^q \in \mathbb{R}^{N_q \times t \times d}$

12:       *# Generate prototypes*

13:       $\mathcal{P}_k \leftarrow \frac{1}{N} \sum_{i=1}^N V_{i,k}^s, \quad \forall k \in \{1, \dots, K\}$     $\triangleright \mathcal{P}_k \in \mathbb{R}^{K \times t \times d}$

14:       *# Compute distances to prototypes and predict labels*

15:       $D_k \leftarrow \text{OTAM}(V^q, \mathcal{P}_k) \quad \forall k \in \{1, \dots, K\}$     $\triangleright D \in \mathbb{R}^{N_q \times K}$

16:       $\hat{Y}^q \leftarrow \arg\max_k D_k$                  $\triangleright \hat{Y}^q \in \mathbb{R}^{N_q}$

17:     **end for**

18: **end for**

19: **Output:** Predicted labels $\hat{Y}^q$ for all query samples

---

**Training.** As described in Algorithm 1, we train our Uni-FSAR model using a prototype learning approach. Before training, we input ground-truth text prompts $\Psi = \{\Psi_c\}_{c=1}^C$ into the UniQ-Former to extract text features $f_\Psi' \in \mathbb{R}^{C \times d}$ (Eq. 7). For each episode, we generate an $N$-shot $K$-way support set $S = \{X^s, Y^s\}$ and query set $Q = \{X^q, Y^q\}$ from the training dataset $\mathcal{D}_{\text{train}}$, and select a $K$-way text feature set $f_\psi' \in \mathbb{R}^{K \times d}$ for the episode's classes from the ground-truth text features $f_\Psi' \in \mathbb{R}^{C \times d}$ (Eq. 7). Both sets are processed by the image encoder to extract image features $F_{\text{img}}^s$ and $F_{\text{img}}^q$, which are then fed into the UniQ-Former along with learned visual queries (VQ) $f_\theta \in \mathbb{R}^{32 \times d}$, learnable text query (LTQ) $f_\phi \in \mathbb{R}^d$, and a uni-directional attention mask $M_{\text{uni}} \in \mathbb{R}^{32 \times 32}$ (Eqs. 3, 4, 5). The UniQ-Former's output (Eq. 5) is processed by the temporal transformer to obtain video features $V^s \in \mathbb{R}^{(N \times K) \times t \times d}$ and $V^q \in \mathbb{R}^{N_q \times t \times d}$. In lines 13–19 of Algorithm 1, we compute the LTQ-based Semantic Bridging (LSB) loss (Eq. 11) to align video features with the target text space. In lines 20–26, we compute the prototype learning loss (Eq. 14) using OTAM (Cao et al., 2020) to assign queries to appropriate class prototypes. The final loss (Eq. 15) is computed in line 28, and model weights are updated in line 29.

**Testing.** During testing, as described in Algorithm 2, we generate episodes by sampling $N$-shot $K$-way support sets $S = \{X^s, Y^s\}$ and query sets $Q = \{X^q, Y^q\}$ from the test dataset $\mathcal{D}_{\text{test}}$. Similar to the training process, we extract image features from both the support and query sets using the image encoder, process them through the UniQ-Former, and feed them into the temporal transformer to obtain video features $V^s \in \mathbb{R}^{(N \times K) \times t \times d}$ and $V^q \in \mathbb{R}^{N_q \times t \times d}$. We compute class prototypes from the support set's video features and use OTAM to calculate distances between the prototypes and query features, assigning each query to the class of the closest prototype.

## A.2 IMPLEMENTATION DETAILS

**Hyperparameters.** To ensure fair comparison, we adopt a consistent hyperparameter settings, following prior work (Wang et al., 2024). We uniformly sample 8 frames from each video and resize them to 255×255 pixels. During training, we apply random cropping to obtain images of 224×224 pixels, while during testing, we use center cropping to achieve the same size. For additional data augmentation, we apply only color jittering. The pretrained Q-Former from the BLIP-2 (Li et al., 2023) model and ViT-L/14 as the image encoder are used. The ViT-L/14 is trained and tested in half-precision (FP16) and kept frozen during training. In the Q-Former, the learned visual queries (VQ) and all weights in the VQ-pathway are frozen, while the learnable text query (LTQ) and all weights in the LTQ-pathway are trained. By default, the temporal transformer consists of 2 layers.

Table 9: The implementation details of our proposed Uni-FSAR.

| Dataset | SSV2 Small | HMDB-51 | UCF-101 | Kinetics |
|---|---|---|---|---|
| Optimizer | Adam, Momentum = 0.9, Nesterov = True | | | |
| Max Epoch | 10 | | | |
| Warm up epoch | 1 | | | |
| Batchsize | 4 | | | |
| Frame | 8 | | | |
| Data augmentation | Color jitter, Random crop | | | |
| Learning rate | 5e-5 | 1e-5 | 2e-6 | 1e-5 |
| Warm up learning rate | 2e-5 | 1e-6 | 1e-7 | 1e-6 |
| Train tasks | 30000 | 3000 | 5000 | 5000 |
| Test tasks | 10000 | | | |
| $\alpha$ (Balance term) | 1/1.2 | 1/1.5 | 1/3 | 1/1.5 |

Table 10: Dataset splits and evaluation settings for few-shot action recognition.

| Dataset | #Classes | #Videos | Train/Val/Test Split | Evaluation Setting |
|---|---|---|---|---|
| UCF101 (Soomro et al., 2012) | 101 | 13,320 | 70 / 10 / 21 | 5-way 1/3/5-shot |
| Kinetics (Carreira & Zisserman, 2017) | 100 | 100 per class (=10,000) | 64 / 12 / 24 | 5-way 1/3/5-shot |
| HMDB51 (Kuehne et al., 2011) | 51 | 6,849 | 31 / 10 / 10 | 5-way 1/3/5-shot |
| SSv2-small (Goyal et al., 2017) | 174 | 100 per class (= 17,400) | 64 / 12 / 24 | 5-way 1/3/5-shot |

We use the Adam optimizer with a single warm-up epoch applied consistently across all datasets. As shown in Tab. 9, we applied different learning rates and weight decay values for each dataset. During training, we assigned a varying number of tasks per epoch for each dataset, while during testing, we evaluated the model using 10,000 tasks per dataset. Additionally, the balance term $\alpha$ in the loss function (Eq. 15) was set differently for each dataset.

**Dataset Characteristics and Split.** As shown in Tab. 10, we evaluate our method on five widely-used few-shot action recognition benchmarks: SSV2-small(Goyal et al., 2017) (Something-Something V2), Kinetics-Fewshot subset(Carreira & Zisserman, 2017), HMDB51(Kuehne et al., 2011), and UCF101(Soomro et al., 2012). To ensure fair comparison, we adopt a consistent split protocol for few-shot evaluation, following prior work (Wang et al., 2024; 2023b; Zhu & Yang, 2018).

- **UCF101 :** Consists of 13,320 videos across 101 action categories. View: predominantly third-person. Source: YouTube and web media. Actions: daily/human activities and sports (e.g., playing instruments, sports skills, simple interactions). The dataset is split into 70 classes for training, 10 for validation, and 21 for testing.

- **Kinetics-Fewshot subset :** A subset of 100 classes is selected from the original 400 categories, with 100 videos per class (10,000 total). View: mostly third-person, diverse camera viewpoints. Source: large-scale web video (YouTube). Actions: broad human actions and interactions spanning everyday activities to sports. Classes are divided into 64 for training, 12 for validation, and 24 for testing.

- **HMDB51 :** Contains 6,766 videos covering 51 action categories. View: third-person; many clips are cinematic or consumer video style. Source: movies, YouTube, and other public media. Actions: body-motion–centric actions and facial/body interactions (e.g., laugh, clap, kick, drink). The dataset is split into 31 classes for training, 10 for validation, and 10 for testing.

- **Something-Something v2 :** Comprises 220,847 videos across 174 fine-grained action categories. SSv2-Small samples 100 videos per class, with 64/12/24 classes for train, validation, and test. View: egocentric (first-person), handheld. Source: crowd-sourced short clips collected to match textual templates/prompts. Actions: fine-grained object manipulations (e.g., moving, pushing, pulling, covering/uncovering common objects). Unlike the third-person datasets above, this egocentric setup implies a slight domain shift. Under our definition of frame-level ambiguity (Section 1), SSv2 does not explicitly exhibit such ambiguity; we therefore include it primarily for fairness and completeness in comparison.

Table 11: Top-1 accuracy (%) of Uni-FSAR on HMDB51 and SSv2-Small under 1-shot and 5-shot settings across 5 random seeds. The last row reports the mean ± standard deviation.

| Seed | HMDB51 | | SSv2-Small | |
|---|---|---|---|---|
| | 1-shot | 5-shot | 1-shot | 5-shot |
| 0 (default) | 82.1 | 90.6 | 53.7 | 68.5 |
| 41 | 81.2 | 90.3 | 54.3 | 68.1 |
| 42 | 82.1 | 90.3 | 53.9 | 67.4 |
| 43 | 81.4 | 90.3 | 54.5 | 67.4 |
| 44 | 81.5 | 90.2 | 54.1 | 68.6 |
| 45 | 81.9 | 90.0 | 54.0 | 67.1 |
| Mean ± Std | **81.6 ± 0.42** | **90.3 ± 0.18** | **54.1 ± 0.28** | **67.8 ± 0.63** |

Table 12: Comparison of model complexity and computational cost between CLIP-FSAR and Uni-FSAR.

| Model | Backbone | Total parameters | Learnable parameters | GFLOPS | GPU Mem. | FPS |
|---|---|---|---|---|---|---|
| CLIP-FSAR | ViT-B/16 | 89.34M | 89.34M | 134.96 | 1.09 GB | 15.32 |
| Uni-FSAR | ViT-L/14 | 508.21M | 68.10M | 641.28 | 2.42 GB | 8.31 |

## A.3 STATISTICAL SIGNIFICANCE

**Seed Sensitivity and Reproducibility Analysis.** To assess the statistical robustness of our results, we conducted five independent training runs of Uni-FSAR under the 5-way 1-shot and 5-shot settings, using five different random seeds (including the default). The top-1 accuracy for each seed on HMDB51 and SSv2-small is reported in Tab. 11.

We observe that the performance is stable across seeds, with low standard deviations. These variations primarily reflect the effects of random initialization and sampling in the few-shot evaluation episodes. The mean and standard deviation values are computed using a simple sample mean and unbiased standard deviation (1-sigma). This confirms the reproducibility and statistical reliability of the proposed method.

## A.4 EXPERIMENTS COMPUTE RESOURCES

**Compute Resources.** All experiments were conducted using a local server equipped with 4 NVIDIA V100 GPUs (32GB each). During training, each model used approximately 14GB of GPU memory per process. The environment was configured with PyTorch 1.9.0+cu111, Torchvision 0.10.0+cu111, and CUDA 11.6. Each episode-level training session for Uni-FSAR took approximately 6–8 hours depending on the dataset (e.g., HMDB51 vs. SSv2-Small), while inference was completed within minutes due to batch-level processing and the frozen backbone structure. For fair comparison, all baseline models were also trained under the same compute setting.

In addition to the main experiments reported in the paper, we performed a number of preliminary and ablation studies (e.g., alternative frame sampling strategies, query token configurations), which consumed approximately 1.5× the compute of the final experimental runs. We provide this information to support reproducibility and transparency regarding the resource requirements of our method.

**Computational Cost Analysis.** As summarized in Tab. 12, Uni-FSAR adopts a larger backbone yet maintains training efficiency by freezing it and updating only lightweight components, including the LTQ-pathway in the UniQ-former and the temporal transformer. Although the backbone is ViT-L/14 rather than ViT-B/16, this choice stems from the BLIP-2 framework design, where the Q-former is only available with a ViT-L/14 vision encoder. Nevertheless, the overall architecture remains computationally practical, supporting real-time inference and stable training. In our experimental environment using four V100 GPUs, CLIP-FSAR encountered out-of-memory (OOM) errors during training, whereas Uni-FSAR trained stably—highlighting its practical viability and resource-efficient design compared to full end-to-end tuning approaches.

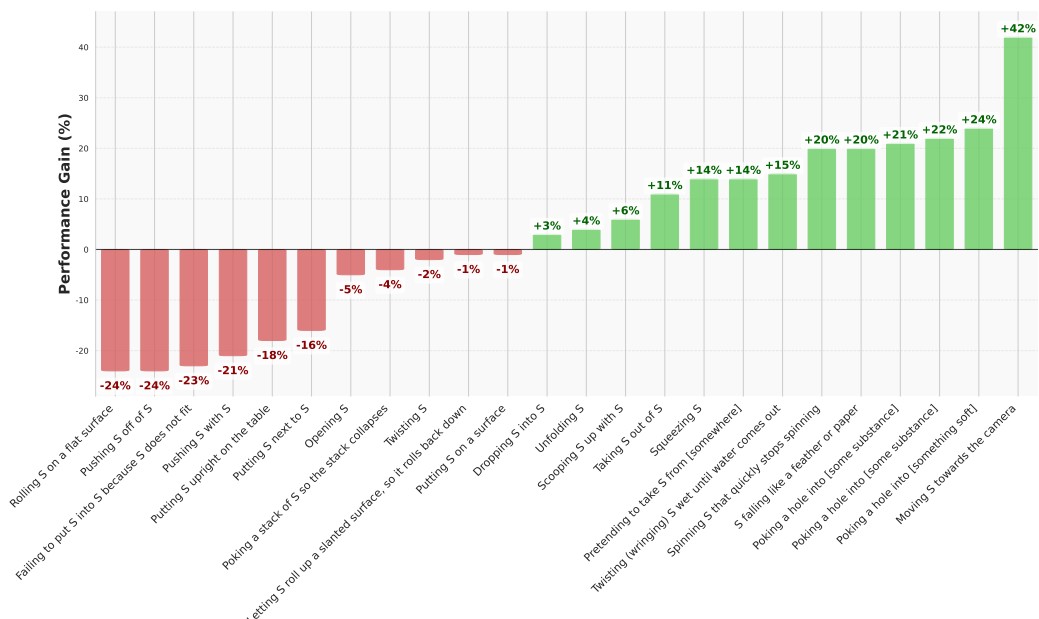

Figure 8: **Class-wise accuracy comparison between Uni-FSAR and CLIP-FSAR on the SSv2-small dataset.** The abbreviation 'S' in the class names stands for 'Something'.

## A.5 MORE QUANTITATIVE RESULTS

**SSv2-small Class-wise Performance.** Figure 8 compares class-wise performance between CLIP-FSAR and the proposed Uni-FSAR on the SSv2-small dataset. Since the official checkpoint for CLIP-FSAR is not available, we reproduced its performance using the same hyperparameters and evaluation settings reported in the original paper (Wang et al., 2024) to ensure a fair comparison. All results are based on the 5-way 1-shot setting.

The SSv2 (Goyal et al., 2017) dataset does not contain frame-level ambiguities as formally defined in our work. In particular, videos do not include irrelevant frames or redundant frames shared across multiple classes. The challenge for this specific dataset, instead lies in aligning fine-grained and continuous actions to complex, compositional class labels.

To address the challenges posed by SSv2, it is essential for a model to (1) accurately capture fine-grained spatial dynamics from limited visual input, and (2) semantically align these observations with long, compositional class labels that often encode complex object relationships and actions. For instance, the class 'Rolling something on a flat surface' requires reasoning over multiple sub-components: the motion primitive "Rolling," the object entities "something" and "flat surface," and the spatial relationship expressed by "on." Given the limited number of examples and frames, this semantic decomposition must be learned from highly constrained supervision.

To our knowledge, such fine-grained label decomposition and alignment has not been explicitly explored in previous few-shot video recognition studies. We identify this as a promising direction for future work, and this is more specifically discussed in Section 5.

We observe that performance drops in some classes in Fig. 8 can be attributed to the above limitations. In contrast, classes showing significant performance gains often contain visually prominent objects, facilitating stronger object–label correspondence. We hypothesize that the superior performance of Uni-FSAR in such classes is due to its improved ability to learn discriminative visual–textual alignments under limited supervision.

**Backbone Fairness Analysis** To address potential concerns regarding backbone fairness, we provide a detailed analysis and controlled experiments demonstrating that the performance gains of Uni-FSAR stem primarily from our novel methodological components (UniQ-Former, uni-directional blending,

Table 13: Proxy and Comparable Backbone Variants for Fairness Analysis (1-shot). Gains are relative to the previous row in each group.

| Variant | HMDB51 | SSv2-Small |
|---|---|---|
| ResNet-50 (OTAM-equivalent) | 54.5 | 36.4 |
| CLIP ViT-B (Freeze) | 58.2 (+3.7) | 29.5 (-6.9) |
| BLIP ViT-B (Freeze) | 52.4 | 28.7 |
| BLIP ViT-B (Uni-Blend + LSB, w/o LTQ) | 53.7 (+1.3) | 32.2 (+3.5) |
| BLIP ViT-B (Uni-Blend + LSB + LTQ) | 58.4 (+4.7) | 33.3 (+1.1) |
| BLIPv2 ViT-L (Vanilla, w/ Q-Former) | 67.0 | 40.5 |
| BLIPv2 ViT-L (Ours: w/ UniQ-Former, Uni-Blend + LSB + LTQ) | 82.3 (+15.3) | 54.1 (+13.6) |

LTQ, and LSB loss) rather than differences in backbone scale. Our choice of BLIPv2 ViT-L was motivated by the need for a Q-Former architecture to support caption-level grounding and selective multi-modal alignment, which is integral to our framework and not feasible in uni-modal backbones like ResNet-50 or simpler multi-modal setups without Q-Former (e.g., CLIP ViT-B in prior works).

**Controlled Backbone Scaling Experiments** We further analyze the results in Table 13 to derive key insights on backbone fairness. The table is divided into three groups for systematic comparison:

*Group 1 (Uni-modal vs. Simple Multi-modal Baseline)*: Starting with ResNet-50 (a uni-modal backbone equivalent to OTAM), switching to frozen CLIP ViT-B yields a modest gain on HMDB51 (+3.7%) but a decline on SSv2-Small (-6.9%). This highlights that simply adopting a pre-trained multi-modal backbone without tailored adaptations does not guarantee consistent improvements, and dataset-specific sensitivities (e.g., temporal reasoning in SSv2) may lead to performance drops.

*Group 2 (BLIP ViT-B Ablation)*: Using frozen BLIP ViT-B as a baseline, incorporating uni-directional blending (Uni-Blend) and LSB loss results in gains of +1.3% on HMDB51 and +3.5% on SSv2-Small. Further adding LTQ boosts performance significantly (+4.7% and +1.1%, respectively), demonstrating the incremental value of our components even on a smaller-scale backbone. Overall, from the frozen baseline, our methods achieve +6.0% on HMDB51 and +4.6% on SSv2-Small, underscoring their effectiveness independent of backbone size.

*Group 3 (BLIPv2 ViT-L with Our Components)*: The vanilla BLIPv2 ViT-L (with standard Q-Former) serves as a strong baseline. Applying our full suite (UniQ-Former, Uni-Blend, LSB, and LTQ) yields substantial gains: +15.3% on HMDB51 and +13.6% on SSv2-Small. These improvements are notably larger than those from mere backbone scaling (e.g., compare to Group 1's mixed results), confirming that our innovations drive the primary performance uplift rather than the larger ViT-L architecture alone.

In summary, across variants, our methodological contributions consistently enhance performance, often outweighing backbone differences. For instance, the gains from our components on BLIPv2 ViT-L (+15.3% and +13.6%) far exceed those from shifting to larger backbones without them. This analysis mitigates fairness concerns by isolating the impact of our novel elements, ensuring the reported advancements are attributable to Uni-FSAR's core design rather than extrinsic factors like model scale. Future work could extend this to even more diverse backbones for broader validation.

**Hyperparameter Sensitivity ($\alpha$).** To verify the stability of the loss balancing term $\alpha$ in Eq. 15, we conducted a sensitivity analysis on HMDB51 and SSv2-Small (5-way 1-shot). We evaluated performance with $\alpha \in \{1.0, 1.5, 3.0\}$ while keeping other settings fixed. As shown in Table 14, the accuracy fluctuates only by $\pm 0.2\%$ on HMDB51 and $\pm 0.5\%$ on SSv2-Small. This indicates that Uni-FSAR is robust to reasonable changes in $\alpha$, and the reported performance does not rely on exhaustive, dataset-specific tuning.

**Feature Space Analysis for Ambiguity Handling.** To quantitatively verify how Uni-FSAR handles frame-level ambiguity without an explicit variance loss, we analyzed the feature statistics on HMDB51. Comparing the pre-trained features to our trained features, we observed that the **frame-level feature variance increased by +302.6%** and the **mean pairwise feature distance increased by +123.0%**. This indicates that our method implicitly learns to disperse frame embeddings in a discriminative

Table 14: Sensitivity analysis of the loss weight $\alpha$. The performance remains stable across different $\alpha$ values.

| $\alpha$ | HMDB51 (1-shot) | SSv2-Small (1-shot) |
|---|---|---|
| 1.0 | 82.1 | 53.3 |
| 1.5 | 82.0 | 53.8 |
| 3.0 | 81.9 | 53.6 |
| **Variation ($\Delta$)** | **$\pm$ 0.2** | **$\pm$ 0.5** |

Table 15: Ablation study on prototype construction strategy: Original (all frames) vs. Top-$K$-only frames.

| Dataset | Shot | Original Acc (%) | Top-$K$-only Acc (%) | $\Delta$ (%) |
|---|---|---|---|---|
| HMDB51 | 1 | 82.3 | 65.5 | -16.8 |
| HMDB51 | 5 | 90.5 | 76.8 | -13.7 |
| SSv2-small | 1 | 54.1 | 41.4 | -12.7 |
| SSv2-small | 5 | 65.2 | 52.3 | -12.9 |

manner, effectively separating informative frames from redundant noise, while maintaining high temporal consistency (cosine similarity $> 0.9$) between adjacent frames.

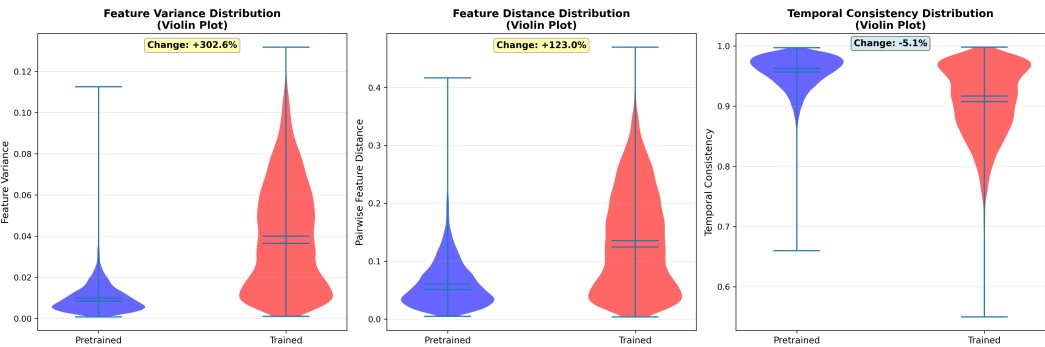

Figure 9: Feature-level statistics before and after training on HMDB51. Uni-FSAR substantially increases feature variance and pairwise feature distance, while keeping temporal consistency high, indicating that frames become more discriminative in the embedding space without introducing excessive noise.

Fig. 9 summarizes as violin plots. The red distributions (trained) are clearly shifted towards larger variance and distance compared to the blue ones (pretrained), while the temporal-consistency distribution remains concentrated near 1.0 with only a modest shift. This visualizes that Uni-FSAR learns to *spread out* frame embeddings in a class-discriminative way without destroying temporal coherence.

These changes in feature-space variance occur together with the **Top-1 accuracy gains on HMDB51** reported in Tab. 1, where frame-level ambiguity is particularly severe. This suggests that the learned increase in inter-frame dispersion is **beneficial**, helping the model focus on truly informative frames and reduce the impact of redundant/irrelevant ones, and ultimately resolve frame-level ambiguity by de-emphasizing noisy content while amplifying distinctive action cues.

Moreover, the same training strategy also improves performance on SSv2-small, where videos are temporally well trimmed and exhibit far fewer irrelevant/redundant frames, indicating that this implicit handling is **not overfitted to a specific dataset** and helps generalization.

**Effectiveness of Top-K Training vs. Inference.**

As shown in Tab. 15, we further investigated the role of Top-K selection. We evaluated a variant that uses only Top-K frames for prototype construction at inference ("Top-K Only Prototype").

This variant degraded performance by approximately **12-17%** across datasets compared to our full model which averages all frames. This confirms our design choice: Top-K acts as a crucial *gradient selector* during training to mitigate noise, while aggregating all frames at inference preserves essential temporal context.

### A.6    MORE QUALITATIVE RESULTS

In addition to the examples presented in Sec. 4.4, we present additional frame-level heatmap visualizations and t-SNE visualization on the HMDB51 dataset to further illustrate the prediction patterns of our proposed model.

**Comparative Analysis of Embedding Distribution.** To qualitatively analyze the discriminative capacity of the learned features, we visualize the class-wise feature embeddings on the HMDB51 test set using t-SNE, as shown in Fig. 10. The top figure corresponds to CLIP-FSAR, while the bottom shows Uni-FSAR. In the CLIP-FSAR embedding space, most classes are densely packed and exhibit significant overlap, indicating limited separation between semantically distinct actions. This suggests that the model struggles to learn discriminative features under few-shot conditions, possibly due to frame-level noise or suboptimal supervision.

In contrast, Uni-FSAR produces clearly separable and compact clusters, especially for classes such as 'kick ball', 'smoke', and 'pour', which exhibit tighter intra-class distributions and lower inter-class confusion. This reflects the model's ability to focus on informative frames via the proposed mechanism, thereby enhancing feature distinctiveness even with limited training data. Overall, the t-SNE visualization supports the quantitative results and further demonstrates that Uni-FSAR better captures class-specific semantics in the few-shot regime compared to CLIP-FSAR.

**Examples of Inter-class Ambiguity.** Figure 11 shows the heatmap for a sample from the 'laugh' class. The baseline model, CLIP-FSAR (Wang et al., 2024), incorrectly predicts the action as 'drink' during frames 0–1 and exhibits a generally noisy confidence distribution, assigning relatively low confidence to the ground-truth class 'laugh'. In contrast, the proposed Uni-FSAR assigns high confidence to 'laugh' specifically in frames 4–7, while appropriately assigning moderate confidence to the 'smoke' class in frames 0–3, where the subject holds a pipe. Figure 12 presents an example from the 'smoke' class. While the baseline incorrectly predicts 'drink' in frames 4–6, our model accurately classifies the action as 'smoke' with high confidence at frame 7. These examples demonstrate that our model outperforms the baseline in addressing the challenges of *inter-class ambiguity* and redundant frames.

**Example of Intra-class Ambiguity.** Figure 13 shows the heatmap for a 'run' class sample. The baseline model misclassifies frame 0 as 'kiss' and frames 1–7 as 'walk', thus failing to correctly identify the ground-truth class 'run'. In contrast, Uni-FSAR eliminates the spurious high confidence for 'kiss' at frame 0 and assigns confidence to both 'walk' and especially 'run' in frames 1–7. This example highlights the superiority of our model in handling *intra-class ambiguity* and irrelevant frames compared to the baseline.

**Examples of Visually Consistent Content.** Figure 14 illustrates the heatmap for a 'pushup' class sample. The baseline incorrectly predicts 'handstand' in frames 0–1 and exhibits low confidence for the correct class 'pushup'. In contrast, Uni-FSAR suppresses the noise in early frames and consistently predicts 'pushup' across all frames. Similarly, Fig. 15 presents a 'fencing' class sample where the baseline shows relatively weak confidence in the ground-truth class compared to our model. Figure 16 illustrates a 'pour' class sample, where CLIP-FSAR shows dispersed and inconsistent predictions, frequently misclassifying frames as 'sword'-related actions. In contrast, Uni-FSAR produces focused and stable predictions aligned with the correct class across all frames, demonstrating improved ability to interpret the visual representation of the given frames accurately. These results show that our model also achieves robust performance on sequences with consistent visual content.

**Qualitative ablation of attention direction.** Each heatmap in Figure 17 visualizes **frame–class attention weights** for the same HMDB51 episode (8 frames, all classes), where the vertical axis denotes the frame index and the horizontal axis denotes the class token. Warmer colors correspond to higher attention between a frame and a class.

**Bi-directional attention.** In the bi-directional design, attention is widely spread across many classes and frames. **Several non-target classes such as *kiss*, *sword* and *flic-flac*** receive relatively strong responses across multiple frames, while the true class *laugh* does not stand out with a clear, concentrated stripe. This pattern indicates that the bi-directional interaction tends to mix visual and textual information symmetrically, but also allows irrelevant classes to keep non-negligible attention, which is consistent with the risk of prototype contamination discussed in the paper.

**Reverse-directional attention.** When we flip the direction (text $\rightarrow$ frames only), the model **fails** to learn a stable alignment: **the heatmap shows strong activations on incorrect classes (again, *chew*, *kiss*, *pour*, *smoke*, *wave*), while the true *laugh* class** receives no distinctive peak. Attention is also concentrated on a few early frames without meaningful differentiation across the rest. This qualitatively matches the severe performance drop ($\sim 20\%$ Top-1) observed for the reverse-directional variant and suggests that pushing information from class tokens back into frame tokens alone is not sufficient to learn reliable prototypes.

**Uni-directional attention (ours).** In contrast, the uni-directional blending (frames $\rightarrow$ text queries) produces a **sharply focused pattern: attention is strongly concentrated on the *laugh* class and *smoke* class**, while other classes remain close to zero across all frames. This matches our Top-$K$ design, where only a few semantically most relevant frames are emphasized, and explains why the uni-directional variant achieves the best performance (82.3% Top-1 vs. 81.2% for bi-directional and $\sim 20\%$ for reverse-directional) despite sharing the same LTQ, LSB, and Top-$K$ components.

The heatmaps thus provide qualitative evidence that uni-directional blending more effectively suppresses cross-class interference and isolates truly informative frame–class interactions, and we observe the same tendency consistently across additional qualitative examples in Figs. 10–16

### A.7 THE USE OF LARGE LANGUAGE MODELS (LLMS)

We used an LLM only for minor grammar and phrasing corrections. It did not contribute to ideas, methods, experiments, analyses, or substantive writing. The authors take full responsibility for all content.

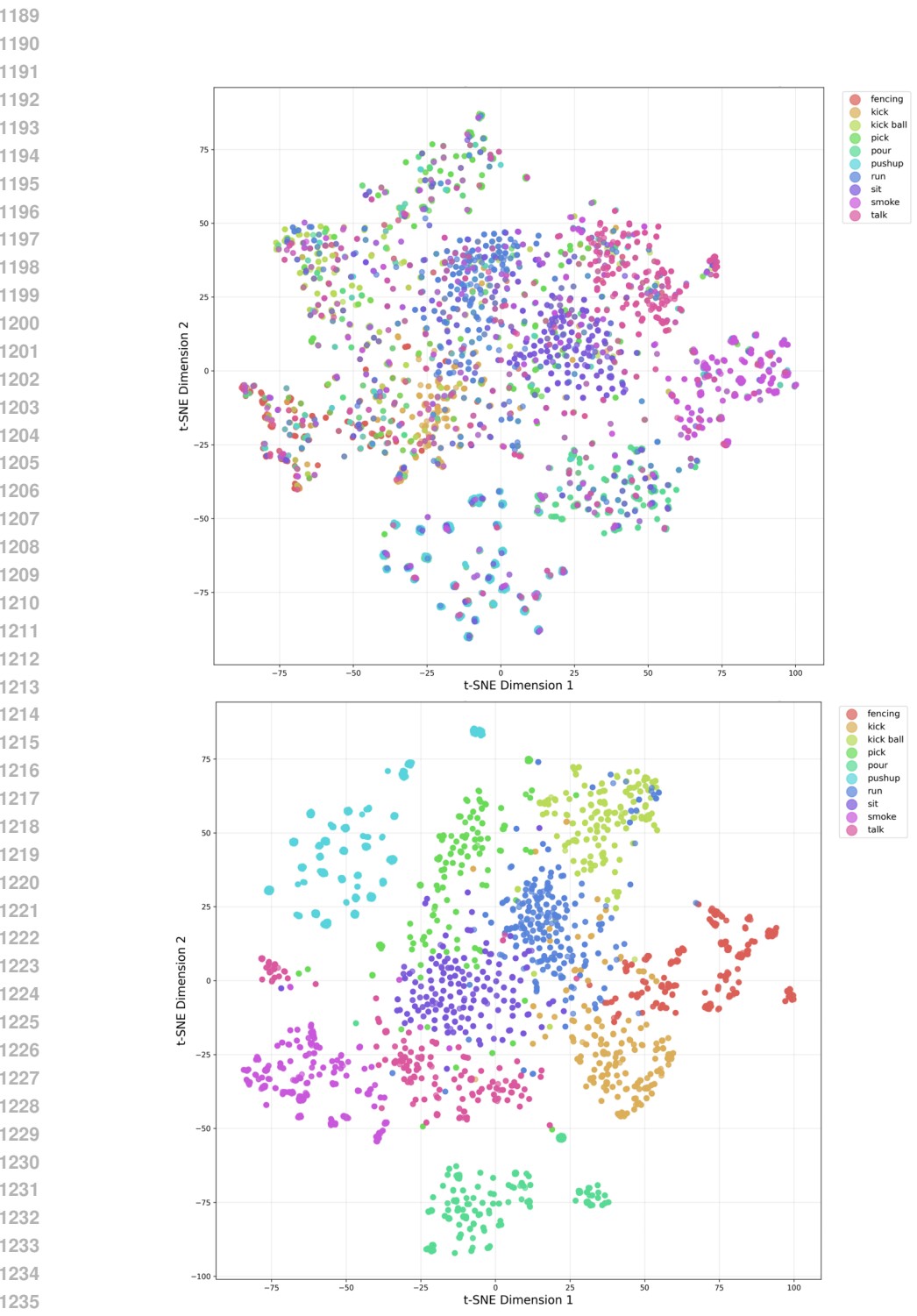

Figure 10: **Comparison between CLIP-FSAR (top) and Uni-FSAR (bottom).** t-SNE visualization of class-wise feature embeddings on the HMDB51 test set.

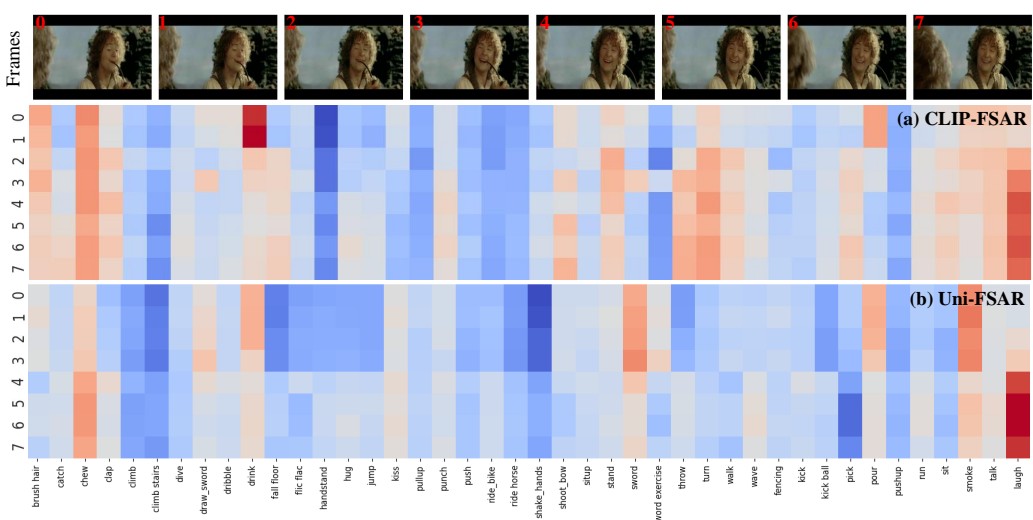

Figure 11: **Examples of frame-level ambiguities in the 'laugh' class.** Heatmap comparison between CLIP-FSAR (a) and Uni-FSAR (b) on an HMDB51 sample. CLIP-FSAR shows noisy activations and high confidence for incorrect classes like 'drink', while Uni-FSAR accurately focuses on 'laugh' in frames 4–7 and reasonably attends to 'smoke' in frames 0–3, demonstrating improved robustness to inter-class ambiguity.

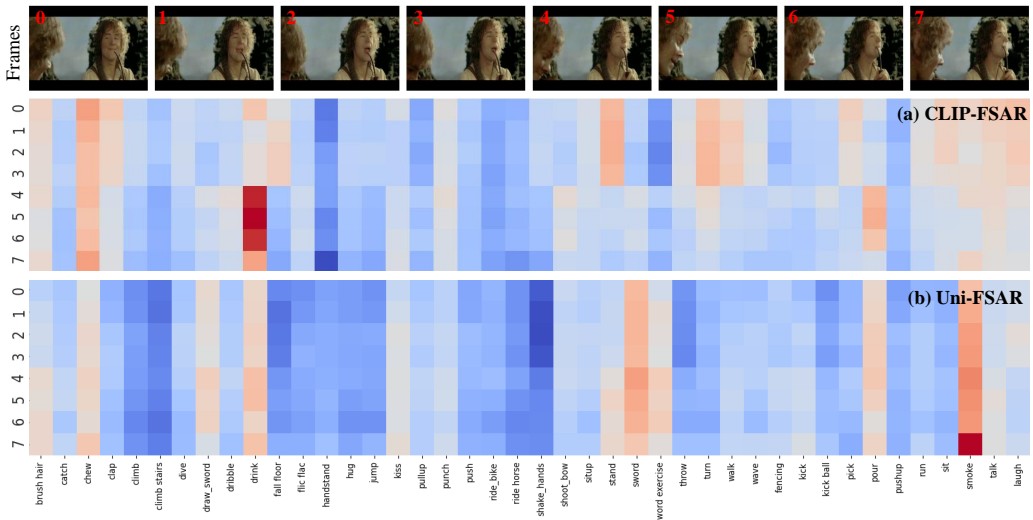

Figure 12: **Examples of frame-level ambiguities in the 'smoke' class.** Comparison of heatmap visualizations between CLIP-FSAR (a) and the proposed Uni-FSAR (b) on a video sample from HMDB51. CLIP-FSAR incorrectly predicts the action as 'drink' in frames 4–6 and fails to assign strong confidence to the correct class 'smoke'. In contrast, Uni-FSAR accurately classifies the action as 'smoke' with high confidence at frame 7, demonstrating its effectiveness in resolving inter-class ambiguity.

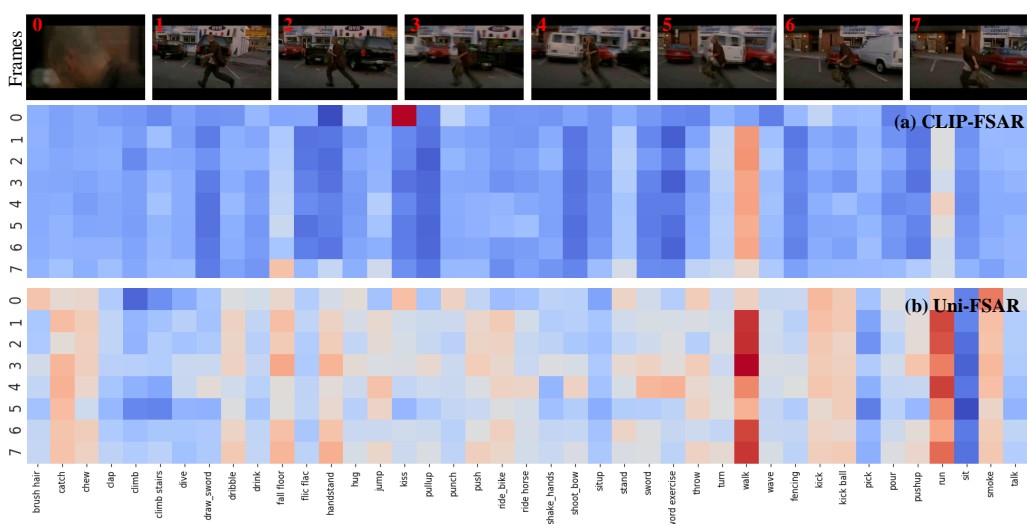

Figure 13: **Examples of frame-level ambiguities in the 'run' class.** Comparison of heatmap visualizations between CLIP-FSAR (a) and the proposed Uni-FSAR (b) on a video sample from HMDB51. CLIP-FSAR misclassifies frame 0 as 'kiss' and frames 1–7 as 'walk', showing confusion across similar motion patterns. Uni-FSAR suppresses the spurious activation at frame 0 and correctly attends to 'run' in frames 1–7. This example illustrates Uni-FSAR's superiority in handling intra-class ambiguity and filtering out irrelevant frames.

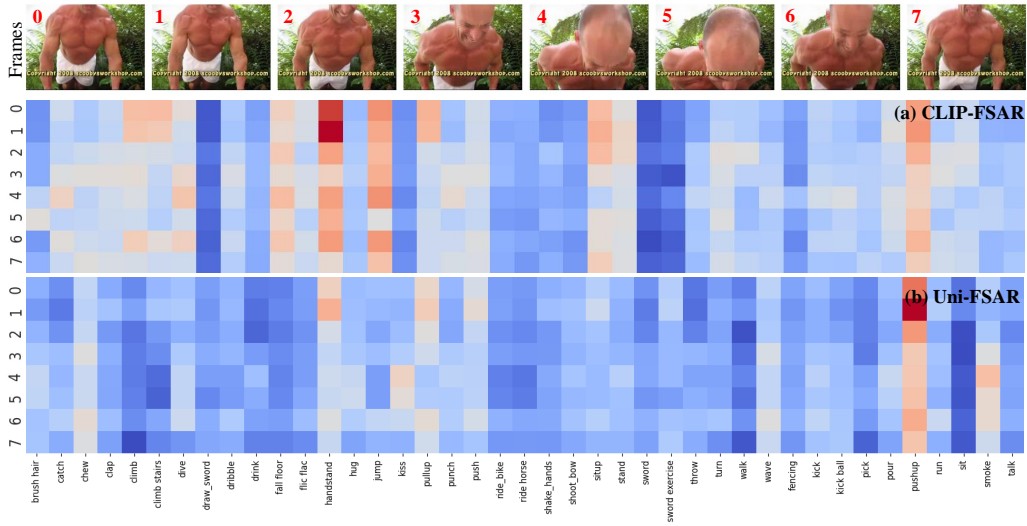

Figure 14: **Examples of visually consistent frames in the 'pushup' class.** Comparison of heatmap visualizations between CLIP-FSAR (a) and the proposed Uni-FSAR (b) on a video sample from HMDB51. CLIP-FSAR predicts 'handstand' in early frames (0–1) and assigns weak confidence to the correct class 'pushup'. In contrast, Uni-FSAR eliminates early-frame noise and consistently predicts 'pushup' across all frames, showing improved stability on consistent content.

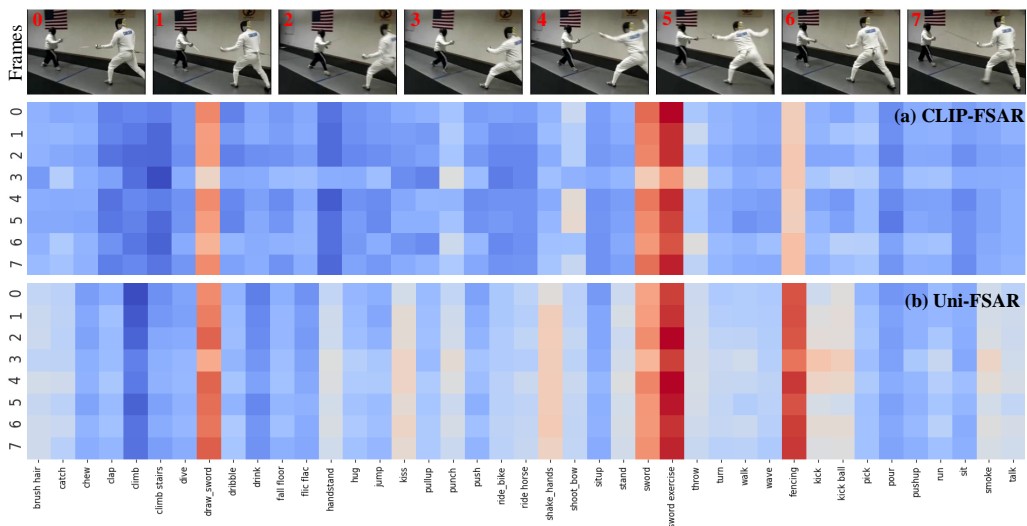

Figure 15: **Examples of visually consistent frames in the 'fencing' class.** Comparison of heatmap visualizations between CLIP-FSAR (a) and the proposed Uni-FSAR (b) on a video sample from HMDB51. CLIP-FSAR assigns weaker confidence to correct class 'fencing' throughout the sequence. In contrast, Uni-FSAR demonstrates more consistent and confident predictions across frames, indicating stronger performance on content with stable semantics. For both models, classes semantically related to the ground-truth label include 'draw sword', 'sword', and 'sword exercise'.

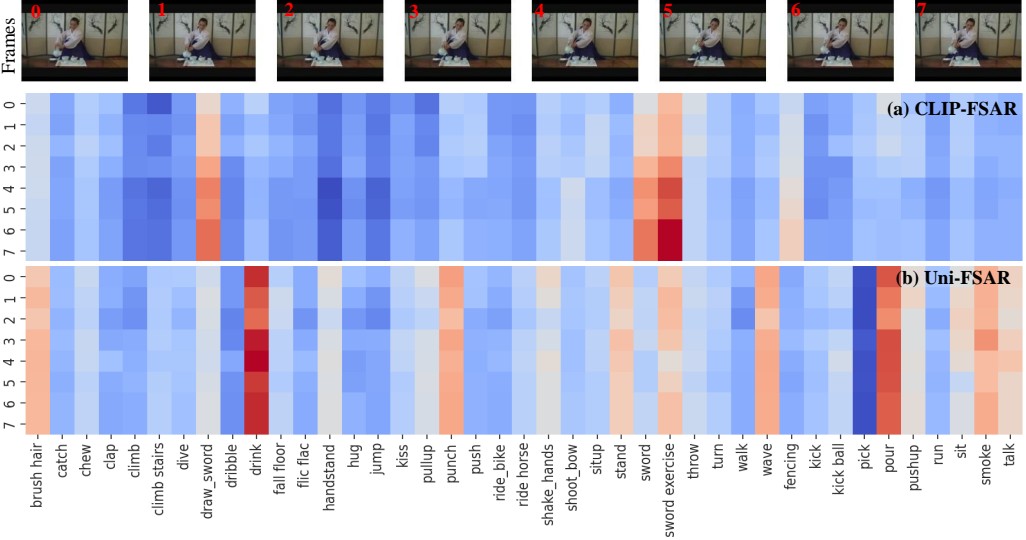

Figure 16: **Examples of visually consistent frames in the 'pour' class.** Comparison of heatmap visualizations between CLIP-FSAR (a) and the proposed Uni-FSAR (b) on a video sample from HMDB51. CLIP-FSAR exhibits inconsistent predictions across frames, with high activations for incorrect classes such as 'draw sword', 'sword', and 'sword exercise', indicating confusion in visual representation. In contrast, Uni-FSAR shows more stable and concentrated responses, correctly attending to action-relevant frames and assigning consistent predictions to the ground-truth classes 'drink' and 'pour'.

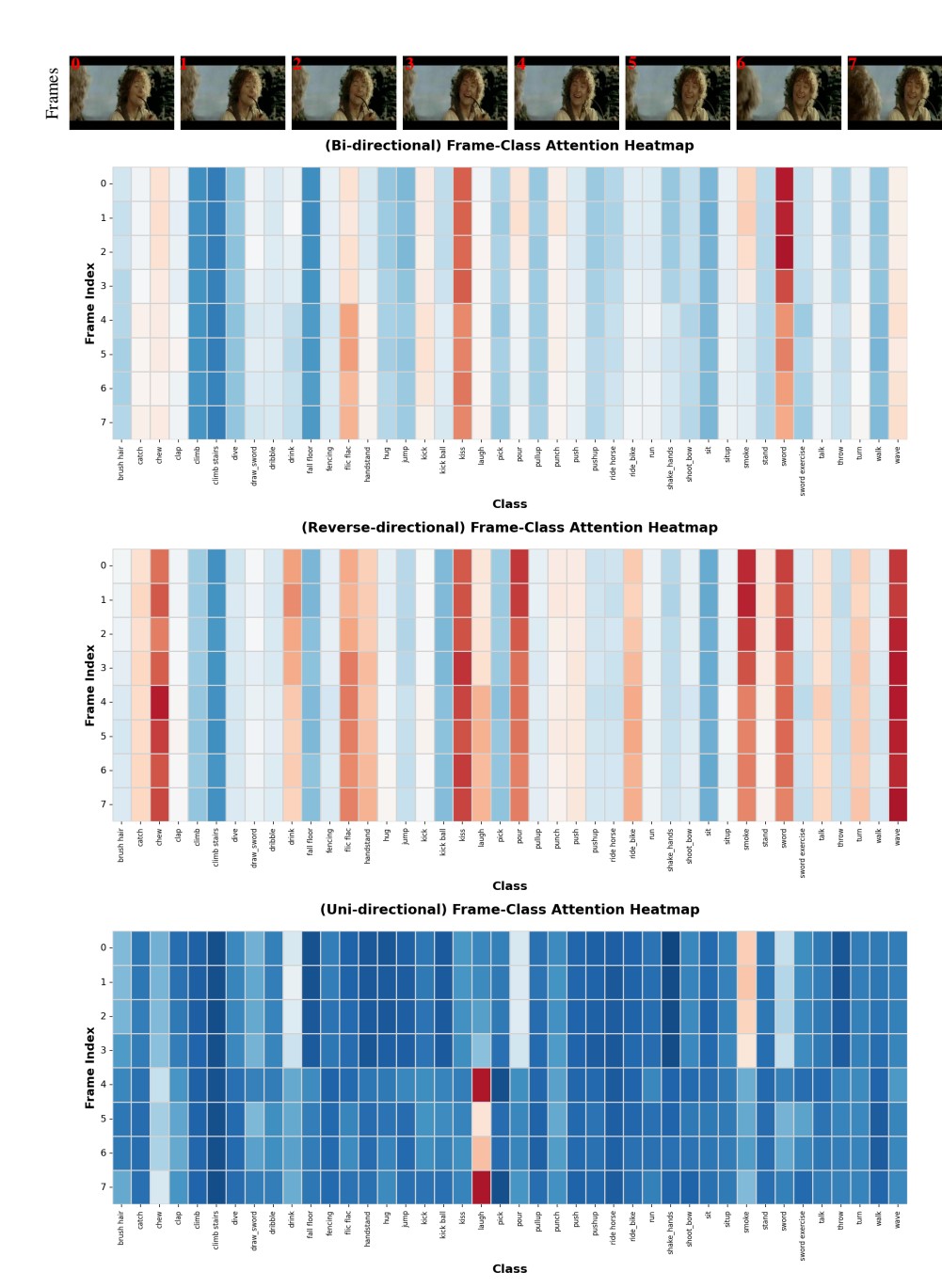

Figure 17: Qualitative comparison of frame–class attention patterns under different blending directions on HMDB51. From left to right, we visualize (i) bi-directional, (ii) reverse-directional, and (iii) our uni-directional blending for the same 8 sampled frames and action classes. All variants share the same Uni-FSAR pipeline (including LTQ, LSB, and Top-$K$ selection); only the attention direction differs. Bi-directional(1st row) and reverse-directional(2nd row) designs spread attention over many classes and often highlight non-target actions, while our uni-directional blending(3rd row) concentrates high attention on the correct class and a small subset of informative frames, consistent with its higher Top-1 accuracy.

