# OpenReview forum: "Uni-directional Blending: Learning Robust Representations for Few-shot Action Recognition with Frame-level Ambiguities"
_ICLR.cc/2026/Conference — Submitted to ICLR 2026_

### Official Review · Reviewer_6P94 · 2025-10-25

**Soundness:** 2
**Presentation:** 2
**Contribution:** 2
**Rating:** 4
**Confidence:** 4

**Summary:**

The paper proposes Uni-FSAR, which introduces uni-directional blending with a learnable text query to mitigate frame-level noise in few-shot action recognition. A Semantic Bridging Loss selectively optimizes only the most relevant frames, and OTAM-based prototype alignment is used for temporal matching.

**Strengths:**

- S1: The motivation behind Uni-FSAR is well-grounded. Frame-level ambiguity represents a critical and realistic challenge in video-based few-shot action recognition.
- S2: Consistent and significant performance gains are reported across multiple standard benchmarks, indicating strong robustness.
- S3: The motivation is effectively conveyed through well-designed and intuitive visual illustrations.

**Weaknesses:**

- W1: The LTQ and uni-directional blending appear to be minor variations of existing cross-attention or masked attention mechanisms in BLIP-2 and text-guided prototype learning.
- W2: The proposed Top-K selection with a contrastive objective is quite similar to hard attention or selective loss used in prior video FSAR works, such as [1].
- W3: SSv2 5way-1shot results in Tab. 2 are weaker or marginal vs SOTA → contradicts "generalizable under ambiguity" claim.
- W4: While Uni-FSAR aims to mitigate frame-level ambiguity by selectively emphasizing the Top-K frames most aligned with text semantics, it remains unclear whether such single-frame–focused selection sufficiently preserves motion cues that are critical for temporal understanding, especially in datasets like SSv2 where the action is defined by subtle frame-to-frame changes rather than static appearances. By suppressing non-selected frames entirely during optimization, the method may risk discarding essential temporal evidence and oversimplifying the underlying action dynamics that require multi-frame context to recognize.
- W5: Figures lack sufficient explanation (e.g., Figure 2 symbols).

[1] Task-adaptive Spatial-Temporal Video Sampler for Few-shot Action Recognition

**Questions:**

- Q1: While an ablation is provided for K=3 in the Top-K strategy, a more detailed explanation or empirical reasoning could further support this selection.
- Q2: It is unclear why LTQ is necessary when CLIP text encoder already embeds class semantics; what exact semantic gap is being "bridged"?
- Q3:  Prototype formation relies on OTAM, but the paper does not clearly explain whether and how the temporal alignment interacts with the proposed uni-directional blending strategy.

---

> ### Author Response · Authors · 2025-11-20
> **Response to Reviewer 6P94 - Part (1/8)**
>
> Comment: We thank R4 for asking for a clearer justification of the Top-K choice. Below we explain the role of Top-K in LSB, why we use $K{=}3$, and how additional variants support this design.
>
> ---
>
> ### **[R4/Q1] Why Top-K and why K=3?**
>
> **Summary**
>
> * In Uni-FSAR, Top-K is used **only in the LSB loss** to route strong gradients to the most LTQ-aligned (semantically informative) frames, while the final prototype still averages **all** frame embeddings.
> * We set **K=3 once** based on the ablation in **Table 6** and reuse it for all datasets and shot settings. It consistently outperforms both **K=1** and the “no Top-K” (GAP-like) variant.
> * Intuitively, **K=1** is too brittle (overly sensitive to ranking noise), while larger K values gradually behave like GAP+Mean and dilute the benefit of selection. For 8-frame episodes, **K=3** strikes a good balance between selectivity and robustness.
>
> ---
>
> ### **(A) Role of Top-K in LSB and choice of K=3**
>
> In Sec. 3.3, LSB computes similarities between frame embeddings and class text, selects the Top-K frames per class, and backpropagates only through those frames. Thus:
>
> * Highly text-aligned frames receive **strong gradients** and are pulled toward the class semantics.
> * Redundant or off-topic frames receive almost **no gradient from LSB** and have weaker influence on the learned prototypes.
>
> The choice of K controls how aggressively we concentrate supervision:
>
> * **K=1** gives supervision to a single frame and is unstable when the top-ranked frame is slightly mis-localized.
> * As **K approaches the total number of frames (T=8 in our episodes)**, LSB degenerates toward GAP+Mean, and redundant frames again receive similar gradients.
>
> In **Table 6**, LSB(Top-3) consistently outperforms GAP+Mean and LSB(Top-1) on both HMDB51 and SSv2-small (1-/5-shot), so we fix **K=3 globally** and do not re-tune it per dataset. For short 8-frame clips, supervising three frames per episode provided the most reliable trade-off in our experiments.

---

> > ### Author Response · Authors · 2025-11-20
> > **Response to Reviewer 6P94 - Part (2/8)**
> >
> > Comment: We appreciate R4’s question about why LTQ is needed in addition to the existing CLIP/BLIP text encoder. Below we clarify what semantic gap we target and how LTQ + LSB operate on top of the standard text encoder.
> >
> > ---
> >
> > ### **[R4/Q2] Why LTQ is needed beyond CLIP/BLIP text embeddings**
> >
> > **Summary**
> >
> > * The frozen CLIP/BLIP text encoder provides coarse class-level semantics (e.g., “kick ball”, “smoke”), but few-shot action clips contain fine-grained, frame-level cues (pre-action, core action, post-action, background), creating both a label–frame **granularity gap** and a pretraining–target **domain gap**.
> > * LTQ is a learnable text-side query token that interacts with each frame. Through our uni-directional blending and temporal transformer, it pulls frame-specific visual evidence into the same embedding space as the frozen class text, and LSB then aligns only the most text-consistent frames using Top-K.
> > * This LTQ + LSB mechanism  **bridges** from coarse class text to frame-level evidence, which is reflected in more separated t-SNE clusters and more stable attention/predictions than CLIP-FSAR.
> >
> > ---
> >
> > ### **(A) Why the text encoder alone leaves a semantic gap**
> >
> > In both CLIP-FSAR and our BLIP-based setting, class semantics come from a frozen text encoder applied to short labels (e.g., “pour”, “smoke”, “brush hair”). This is powerful but incomplete for FSAR because:
> >
> > * **Granularity mismatch.**
> >   Each video is a sequence of heterogeneous frames: preparation, contact, follow-through, and often irrelevant/background segments. A single short phrase like “smoke” does not specify which frames truly express the action versus which are off-topic. Simply aligning all frame embeddings to one class vector treats these frames as equally relevant.
> >
> > * **Domain mismatch (web images → noisy action videos).**
> >   CLIP/BLIP text embeddings are learned from web-scale image–caption pairs, whereas FSAR operates on short, noisy clips with different statistics and ambiguity patterns. The class embedding for “fencing” or “brush hair” is not tailored to intra-video dynamics or frame-level ambiguity in HMDB51 / SSv2-small.
> >
> > This is consistent with our analyses in Appendix A.6: t-SNE shows overlapping class clusters for CLIP-FSAR (e.g., *drink*, *smoke*, *brush hair*), and qualitative sequences show prediction flips on off-topic frames, indicating that the coarse class text alone is not enough to organize frame embeddings properly.
> >
> > ---
> >
> > ### **(B) How LTQ + LSB actually bridge the gap**
> >
> > LTQ is introduced so that frame-level evidence can reshape how the class text is realized in the joint embedding space, rather than relying solely on a coarse, word-level label embedding.
> >
> > Concretely, for each frame LTQ is updated via uni-directional blending to absorb fine-grained visual cues such as body pose, object configuration, and which sub-phase of the action is occurring (preparation, contact, follow-through). Over training, this makes LTQ a compact text-side descriptor of “how this class typically looks in this domain,” instead of just reflecting the original short phrase (e.g., *“smoke”*, *“kick ball”*) learned from web captions.
> >
> > LSB then uses similarities between LTQ-based frame descriptors and the frozen class text to drive the alignment:
> >
> > * Frames that genuinely depict the labeled action are repeatedly pulled closer to the class text, so the region around each class embedding is expanded and shaped to cover diverse but correct, fine-grained realizations of that action in the target video domain.
> > * Redundant or off-topic frames are rarely selected in Top-K, receive little gradient from LSB, and therefore remain farther from the class text and have weaker influence on prototypes.
> >
> > In this way, LTQ + LSB do more than **re-weight static text embeddings**: they let frame-wise evidence iteratively refine how each class is represented in feature space, narrowing the label–frame granularity gap and adapting the pre-trained text space to the statistics and ambiguities of few-shot action videos.

---

> ### Author Response · Authors · 2025-11-20
> **Response to Reviewer 6P94 - Part (3/8)**
>
> **Comment.**
> We thank R4 for pointing out the connection between our representation learning pipeline and OTAM-based temporal alignment. Below we clarify where OTAM sits in the architecture and how it interacts with uni-directional blending.
>
> ---
>
> ### **[R4/Q3] Interaction between uni-directional blending and OTAM**
>
> **Summary**
>
> * OTAM is used as a **fixed** temporal alignment module, identical across all variants (ours and baselines); we do not modify its formulation.
> * Uni-directional blending, LTQ, and LSB operate **before** OTAM, reshaping frame-wise features into ambiguity-robust sequences $\(V^s, V^q\)$ that OTAM then aligns.
> * Under the same OTAM setup, turning on our modules yields large gains (e.g., **+15.3% / +13.6%** on HMDB51 / SSv2-small 1-shot), indicating that improvements come from better representations rather than changes to the alignment mechanism.
>
> ---
>
> ### **(A) How Uni-FSAR and OTAM interact**
>
> In Uni-FSAR, all proposed components act at the feature level **before** OTAM:
>
> * **Uni-directional blending + LTQ in UniQ-Former.**
>   Uni-directional blending and LTQ are applied inside UniQ-Former to produce per-frame, text-grounded descriptors that are less affected by ambiguous or off-topic frames.
>
> * **Temporal transformer to form ordered sequences.**
>   A temporal transformer then turns these descriptors into ordered frame sequences $\(V^s, V^q \in \mathbb{R}^{t \times d}\)$ for support and query videos.
>
> * **LSB to bias supervision toward informative frames.**
>   LSB further biases these sequences by giving strong gradients only to Top-K text-aligned frames, so redundant or off-topic frames have weaker influence on the learned features.
>
> OTAM is applied **after** this pipeline, it computes distances along monotonic alignment paths between the cleaned sequences $\(V^s\)$ and $\(V^q\)$ and uses them for prototype-based matching.
>
> In other words, OTAM’s temporal alignment is unchanged; **what changes is the quality of the frame trajectories it receives.** Our method feeds OTAM more discriminative, ambiguity-robust frame sequences rather than raw, noisy embeddings.
>
> ---
>
> ### **(B) Evidence that gains come from representations.**
>
> Because OTAM is kept fixed across all variants (including CLIP-FSAR-style baselines), any performance difference directly reflects improvements in the representations passed into OTAM. In particular, when we turn on uni-directional blending, LTQ, and LSB under the same BLIPv2 ViT-L backbone and OTAM configuration, we observe gains of **+15.3%** (HMDB51 1-shot) and **+13.6%** (SSv2-small 1-shot). This supports our claim that Uni-FSAR primarily enhances how frame sequences are encoded and cleaned **before** temporal alignment, rather than relying on any modification of the OTAM mechanism itself.

---

> > ### Author Response · Authors · 2025-11-20
> > **Response to Reviewer 6P94 - Part (4/8)**
> >
> > Comment: We appreciate R4’s concern that LTQ and uni-directional blending might be minor variations of existing cross-/masked-attention in BLIP-2 or text-guided prototype learning. Below we clarify how these components are specifically re-purposed and extended for frame-level ambiguity in FSAR.
> >
> > ---
> >
> > ### **[R4/W1] LTQ and uni-directional blending vs. existing masked attention**
> >
> > **Summary**
> >
> > * Our core contribution is not generic masked attention, but a FSAR-specific framework of **uni-directional blending** and a **learnable text query (LTQ)** that is trained end-to-end to handle *frame-level ambiguity* (redundant / off-topic frames and overlapping actions).
> > * BLIP-2’s Q-Former masks are designed for ITC/ITM/ITG on static images (contrastive, matching, causal decoding), and do not implement a one-way “visual → text” pathway that explicitly prevents prototype contamination across ambiguous frames.
> > * Unlike prior text-guided prototype approaches that rely on fixed class text embeddings, LTQ is a trainable text-side token updated by video frames and then coupled with LSB and Top-K inside the loss, yielding large FSAR gains (up to 20–30 points in cross-dataset settings) on the **same** backbone.
> >
> > ---
> >
> > ### **(A) How our design differs from BLIP-2 and generic text-guided prototypes**
> >
> > * **Uni-directional blending.**
> >   Prior VLM-FSAR works (e.g., CLIP-FSAR) use bi-directional blending where text and frame tokens freely attend to each other, and BLIP-2 uses task-specific masks (ITC/ITM/ITG) for static image–text objectives. In Uni-FSAR, we introduce a **uni-directional mask** that lets all queries attend to visual tokens but blocks attention into LTQ. LTQ therefore serves as a one-way “reader” of frame evidence, explicitly designed to avoid pushing noisy class-text signals back into frame tokens under frame-level ambiguity.
> >
> > * **LTQ as a trainable text-side bridge.**
> >   Instead of only using frozen class label embeddings, we add a **learnable text query** that is updated by video frames through uni-directional attention and then aligned to the frozen class text via LSB. This differs from standard text-guided prototype learning, where static label embeddings or prompts re-weight visual features but are not themselves adapted to the fine-grained, frame-level structure of FSAR videos.
> >
> > * **Integration with LSB and Top-K.**
> >   LTQ is not an isolated token: it is tightly coupled with the LTQ-based Semantic Bridging (LSB) loss and Top-K selection. LSB uses LTQ–frame similarities to select and supervise only the most label-consistent frames during training, thereby reshaping the embedding space specifically to mitigate intra-/inter-class ambiguity rather than just improving generic image–text alignment.
> >
> > ---
> >
> > ### **(B) Evidence that this is more than a minor variation**
> >
> > All these components are trained on the **same** BLIP-family backbones and OTAM alignment. As shown in our ablations, incrementally enabling uni-directional blending, LTQ, and LSB on a fixed backbone yields substantial improvements (e.g., double-digit gains in 1-shot settings and up to roughly 20–30 point gains in cross-dataset transfer), whereas merely switching masks or backbones without these modules does **not** produce comparable benefits. This indicates that the proposed LTQ + uni-directional blending pathway is a targeted FSAR innovation for frame-level ambiguity, rather than a minor cosmetic change to existing masked attention.

---

> > > ### Author Response · Authors · 2025-11-20
> > > **Response to Reviewer 6P94 - Part (5/8)**
> > >
> > > **Comment:**
> > > Thank you for pointing out [1]. While both methods use a form of Top-K, the role and mechanism of our LTQ-based Semantic Bridging (LSB) loss are fundamentally different from the hard/Top-k attention used in prior CNN-based FSAR.
> > >
> > > ---
> > >
> > > ### **[R4/W2] How LSB differs from prior hard/Top-k attention**
> > >
> > > **Summary**
> > >
> > > * In [1], Top-k is a *sampler*: it selects a subset of raw frames (and regions) **before** the backbone. In Uni-FSAR, Top-K appears only *inside the loss*, **after** VLM-based encoding, as a **text-guided gradient selector**; all methods still use the same 8 frames and the same OTAM alignment.
> > > * The selector in [1] is *text-agnostic*, driven by generic visual importance. Our Top-K is driven by **LTQ–class-text similarity**, explicitly targeting the label–frame semantic gap and frame-level ambiguity.
> > > * [1] is purely CNN-based FSAR. Our LSB is tightly coupled with **LTQ and uni-directional blending on BLIP/BLIP-2**, and, to our knowledge, is the first FSAR objective that selectively aligns only text-consistent frames to class labels in a VLM embedding space.
> > >
> > > ---
> > >
> > > ### **(A) Point-by-point distinction w.r.t. [1]**
> > >
> > > * **Where Top-K acts.**
> > >   In [1], non-selected frames are **dropped** before the main backbone, changing the input sequence. In our case, Top-K is applied to LTQ-based frame embeddings inside LSB; non-selected frames **still** pass through the backbone and OTAM, but receive no gradient from $\(L_{\text{LSB}}\)$. Prototype formation and temporal alignment therefore remain identical across baselines and Uni-FSAR.
> > >
> > > * **What drives selection.**
> > >   The hard/Top-k in [1] is supervised only by classification loss and does **not** know how frames relate to the class *text*. Our Top-K is computed over similarities between LTQ features and frozen class text. Frames that truly express the labeled action pull the LTQ–text prototype to cover fine-grained poses/interactions in the video domain, whereas redundant frames are de-emphasized. This directly addresses the semantic gap defined in Sec. 1 and R4/Q2.
> > >
> > > * **Objective and regime.**
> > >   The main objective of [1] is efficient sampling of long videos under a compute budget. Our setup keeps compute, frames, and OTAM fixed, and uses LSB’s Top-K purely to learn **ambiguity-robust prototypes** on top of a VLM backbone. This VLM+LTQ+LSB design and its goal (frame-level ambiguity reduction rather than sampling efficiency) are **not covered by [1].**
> > >
> > > ---
> > >
> > > In this sense, LSB’s Top-K is not a re-use of existing hard attention, but a text-conditioned, representation-level mechanism tailored to bridging the label–frame semantic gap under frame-level ambiguity in VLM-based FSAR.

---

> > > > ### Author Response · Authors · 2025-11-20
> > > > **Response to Reviewer 6P94 - Part (6/8)**
> > > >
> > > > **Comment:**
> > > > We appreciate this concern and clarify that “generalizable under ambiguity” is defined with respect to *ambiguity-heavy* datasets and cross-dataset transfer. SSv2-small, which is largely free of frame-level ambiguity under our definition, is included as a complementary sanity check rather than the primary evidence for this claim.
> > > >
> > > > ---
> > > >
> > > > ### **[R4/W3] How SSv2-small fits our “generalizable under ambiguity” claim**
> > > >
> > > > **Summary**
> > > >
> > > > * Our claim targets settings where **frame-level ambiguity is prominent** (HMDB51, UCF101, Kinetics) and cross-dataset transfer involving such domains; SSv2-small is explicitly described (App. A.5) as having almost **no frame-level ambiguity.**
> > > > * On SSv2-small 1-shot, Uni-FSAR is essentially on par with CLIP-FSAR (54.1 vs. 54.5), and **clearly stronger in 3-/5-shot**, indicating that our design does not hurt performance on an ambiguity-less but fine-grained dataset.
> > > > * The main empirical support for “generalizable under ambiguity” comes from **large cross-dataset gains** in Tab. 3 when transferring between HMDB51 (ambiguity-heavy) and SSv2-small (ambiguity-less).
> > > >
> > > > ---
> > > >
> > > > ### **(A) Scope of the claim and the role of SSv2-small**
> > > >
> > > > As noted in App. A.5, SSv2-small clips are temporally well trimmed and contain very few irrelevant or redundant frames shared across classes; the difficulty is fine-grained temporal reasoning and compositional labels, not frame-level ambiguity.
> > > >
> > > > In contrast, HMDB51 (and related web-style benchmarks) often include off-topic or redundant segments, which is the primary regime our method targets. Our “generalizable under ambiguity” phrasing thus refers to
> > > > (i) strong results on ambiguity-heavy datasets and
> > > > (ii) robust cross-dataset transfer when training or testing on such domains.
> > > >
> > > > SSv2-small is reported to show that this specialization does **not** degrade performance on an ambiguity-less benchmark.
> > > >
> > > > ---
> > > >
> > > > ### **(B) Reading SSv2-small numbers in Tab. 2**
> > > >
> > > > For SSv2-small (5-way FSAR), Tab. 2 shows:
> > > >
> > > > * **1-shot:** CLIP-FSAR 54.5, Uni-FSAR 54.1 (−0.4 pt, essentially comparable)
> > > > * **3-shot:** CLIP-FSAR 58.6, Uni-FSAR 64.4 (+5.8 pts)
> > > > * **5-shot:** CLIP-FSAR 61.8, Uni-FSAR 68.8 (+7.0 pts)
> > > >
> > > > Moreover, the shot-wise gain is much larger for Uni-FSAR:
> > > >
> > > > * CLIP-FSAR: 54.5 → 61.8 (1 → 5 shot: +7.3 pts)
> > > > * Uni-FSAR: 54.1 → 68.8 (1 → 5 shot: +14.7 pts)
> > > >
> > > > This pattern indicates that, even when frame-level ambiguity is minimal, our representation **better exploits additional clean support evidence**, rather than overfitting to ambiguous frames. The slight 1-shot gap on SSv2-small does not contradict our claim; it shows that we remain competitive in this complementary, ambiguity-less regime.
> > > >
> > > > ---
> > > >
> > > > ### **(C) Cross-dataset evidence for generalizability under ambiguity**
> > > >
> > > > The strongest evidence for our claim comes from the cross-dataset results in Tab. 3:
> > > >
> > > > * **HMDB51 → SSv2-small:**
> > > >   CLIP-FSAR: 33.9 / 46.5 (1/5-shot)
> > > >   Uni-FSAR: 52.3 / 68.2, i.e., **+18.4 / +21.7** points.
> > > >
> > > > * **SSv2-small → HMDB51:**
> > > >   CLIP-FSAR: 37.1 / 46.3
> > > >   Uni-FSAR: 72.7 / 85.0, again with very large margins.
> > > >
> > > > Here, whether we train on an ambiguity-heavy domain (HMDB51) and transfer to a less ambiguous one (SSv2-small), or vice versa, Uni-FSAR consistently outperforms CLIP-FSAR by substantial margins. This is precisely the sense in which we use “generalizable under ambiguity”: representations learned with our ambiguity-focused design transfer more robustly across domains with different levels of frame-level ambiguity.

---

> > > > > ### Author Response · Authors · 2025-11-20
> > > > > **Response to Reviewer 6P94 - Part (7/8)**
> > > > >
> > > > > **Comment:**
> > > > > We clarify that our Top-$K$ mechanism **does not** drop frames or remove their temporal information from the model. All frames are always used for temporal modeling and prototype formation; Top-$K$ only gates gradients from the auxiliary LSB loss. Motion cues and multi-frame context are therefore preserved, as also reflected by our behaviour on SSv2-small.
> > > > >
> > > > > ---
> > > > >
> > > > > ### **[R4/W4] Top-$K$ selection vs. temporal information loss**
> > > > >
> > > > > **Summary**
> > > > >
> > > > > * Top-$K$ is applied **only inside the LSB loss**: non-selected frames still pass through the temporal transformer, contribute to the final prototype, and are aligned by OTAM.
> > > > > * Thus, motion cues are preserved at the representation and metric levels; Top-$K$ merely biases supervision toward semantically informative frames instead of enforcing single-frame decisions.
> > > > > * On SSv2-small, where actions hinge on subtle temporal changes, Uni-FSAR remains comparable to CLIP-FSAR in 1-shot and clearly better in 3-/5-shot, indicating that temporal evidence is not discarded.
> > > > >
> > > > > ---
> > > > >
> > > > > ### **(A) What Top-$K$ actually does (and does not do)**
> > > > >
> > > > > As defined in Sec. 3.3, we compute a similarity matrix between class text and frame embeddings, select the Top-$K$ frames, and apply the LSB loss only on those frames. Importantly:
> > > > >
> > > > > * Only Top-$K$ frames receive gradients from $L_{\text{LSB}}$,
> > > > > * **All** frames still receive gradients from the base FSAR loss $L_{\text{FSAR}}$,
> > > > > * **All** frames are included when forming the final video prototype via averaging (Eq. (12)), which OTAM uses at inference (Sec. 3.4).
> > > > >
> > > > > Non-selected frames are therefore **not removed** from the temporal pipeline. They remain part of the sequence processed by the temporal transformer and appear in the prototypes that OTAM aligns. Top-$K$ only serves as a semantic supervision mask for the auxiliary text-alignment objective.
> > > > >
> > > > > ---
> > > > >
> > > > > ### **(B) Why motion cues and multi-frame context are preserved**
> > > > >
> > > > > Temporal information is handled independently of Top-$K$:
> > > > >
> > > > > * The temporal transformer $T(\cdot)$ (Sec. 3.2) operates on the **full** sequence of frame embeddings, learning inter-frame relations across all $T$ frames, regardless of whether they are selected by LSB.
> > > > > * OTAM (Sec. 3.4) then performs ordered alignment over these full sequences, with alignment paths free to traverse or skip any time step. Since prototypes are built from all frames, OTAM still sees the complete temporal evolution of each clip.
> > > > >
> > > > > In this design, Top-$K$ introduces a semantic bias during training (stronger supervision on frames that best match the label), but it does **not** constrain the model to single-frame recognition at inference. Motion patterns that require multi-frame context, including the subtle frame-to-frame changes characteristic of SSv2, remain available to both $T(\cdot)$ and OTAM.
> > > > >
> > > > > ---
> > > > >
> > > > > ### **(C) Evidence from SSv2-small and qualitative behaviour**
> > > > >
> > > > > If Top-$K$ were discarding essential temporal evidence, we would expect a clear degradation on SSv2-small, where actions are defined by fine temporal changes. However, Tab. 2 shows:
> > > > >
> > > > > * **1-shot:** CLIP-FSAR 54.5 vs. Uni-FSAR 54.1 (essentially tied),
> > > > > * **3-/5-shot:** Uni-FSAR clearly outperforms CLIP-FSAR (64.4 vs. 58.6, and 68.8 vs. 61.8).
> > > > >
> > > > > Moreover, the improvement from 1 → 5 shots is larger for Uni-FSAR than for CLIP-FSAR, indicating that our representation can exploit additional clean temporal evidence rather than oversimplifying it.
> > > > >
> > > > > Qualitative sequences (Sec. 4.4, Fig. 7; App. A.6, Figs. 10–15) also show that Uni-FSAR maintains coherent multi-frame reasoning: it focuses on action-relevant segments over time and yields more stable predictions, while CLIP-FSAR often fluctuates on off-topic frames. This is consistent with our intent: use Top-$K$ to reduce the influence of ambiguous frames, **without** discarding the temporal structure needed for action understanding.
> > > > >
> > > > > ---

---

> > > > > > ### Author Response · Authors · 2025-11-20
> > > > > > **Response to Reviewer 6P94 - Part (8/8)**
> > > > > >
> > > > > > ### **[R4/W5] Figures lack sufficient explanation (e.g., Figure 2 symbols).**
> > > > > > **Response.**
> > > > > > We agree that several figures can be made clearer and will improve their exposition in the revised version. These changes are purely presentational and do not affect the method or results.
> > > > > >
> > > > > > * We will add more detailed captions and legends for all figures, explicitly explaining symbols, arrows, and color/shape codes, and ensuring that each visual element is referenced at least once in the main text.
> > > > > > * For Fig. 2 in particular, we will clarify the meaning of all icons and symbols (e.g., LTQ vs. visual queries, the uni-directional attention mask, OTAM alignment paths, and different frame types) so that the interaction among modules can be understood without guessing.
> > > > > >
> > > > > > These edits will improve readability and help readers more easily follow how Uni-FSAR, LTQ, Top-$K$ selection, and OTAM interact in the overall pipeline.

---

### Official Review · Reviewer_1jCD · 2025-10-30

**Soundness:** 2
**Presentation:** 2
**Contribution:** 2
**Rating:** 4
**Confidence:** 4

**Summary:**

This paper introduces the Uni-FSAR framework. The authors identify two types of ambiguities: (1) intra-class ambiguity from irrelevant frames within the same action class, and (2) inter-class ambiguity from redundant frames shared across classes. To tackle these challenges, three core components are introduced: uni-directional blending strategy, Learnable Text Query, and LTQ-based Semantic Bridging Loss.

**Strengths:**

1) Problem formulation sharply defines intra- and inter-class frame-level ambiguities with quantitative evidence, providing a compelling, data-driven motivation.
2) Extensive experiments demonstrate SOTA performance across multiple benchmarks, with particularly impressive gains on noisy datasets and strong cross-dataset generalization capabilities.

**Weaknesses:**

1) The paper compares Uni-FSAR using BLIPv2 ViT-L/14 (508.21M total parameters) against CLIP-FSAR using ViT-B/16 (89.34M parameters). This 5.7× difference in total parameters and the substantially larger vision encoder make it impossible to isolate whether the reported improvements stem from the proposed methodological innovations or simply from using a more powerful backbone. (Tables 1-2)
2) The proposed method directly applies the Q-Former with 32 learnable visual queries originally designed for static images to each video frame. While this reuse simplifies integration, it remains unclear whether these static-image queries are sufficient to capture temporal dynamics or motion-related cues that are crucial for video-based action understanding.
3) The paper's core components—uni-directional attention masking, learnable text queries, and Top-K frame selection—are well-established techniques borrowed from existing work.

**Questions:**

Please refer to weakness.

---

> ### Author Response · Authors · 2025-11-20
> **Response to Reviewer 1jCD - Part (1/3)**
>
> Comment: We appreciate R3’s concern that the 5.7× parameter gap between BLIPv2 ViT-L/14 and CLIP ViT-B/16 may obscure whether improvements come from our method or from a stronger backbone. Below we explain how Table 1 is designed to disentangle these factors and why its key numbers indicate that the dominant gains come from Uni-FSAR, not from backbone scaling.
>
> ---
>
> ### **[R3/W1] Backbone scale vs. methodological contribution**
>
> **Summary**
>
> * To address fairness, we perform controlled analyses where the backbone is **fixed** and we only add Uni-Blend, LTQ, and LSB (Table 1). On the lighter BLIP ViT-B backbone (closer in capacity to CLIP-FSAR), our modules bring 1-shot gains of **+6.0** points on HMDB51 and **+4.6** points on SSv2-small *within the same backbone*.
> * On BLIPv2 ViT-L (the 508M-parameter backbone flagged by the reviewer), adding UniQ-Former, Uni-Blend, LTQ, and LSB yields additional 1-shot gains of **+15.3** points on HMDB51 and **+13.6** points on SSv2-small, again with the backbone fixed. These margins are far larger than typical backbone-only swaps and cannot be explained by capacity increase alone.
> * Moreover, the frozen CLIP ViT-B baseline still outperforms frozen BLIP ViT-B by about **+5.8** points on HMDB51 and **+0.8** points on SSv2-small, showing that the BLIP variants are *not* inherently stronger than CLIP in our setting. The substantial gains observed on BLIP backbones therefore mainly reflect the effect of Uni-Blend, LTQ, and LSB, rather than a privileged choice of a stronger vision encoder.
>
> ---
>
> For clarity, we reproduce the key fairness table used in our analysis.
>
> **Table 1: Proxy and comparable backbone variants for fairness analysis (1-shot).**
> We fix the backbone within each BLIP family and incrementally add our components.
>
> | Backbone               | QFormer | Uni-Blend | LTQ | LSB | HMDB51 | SSv2-small |
> |------------------------|:-------:|:---------:|:---:|:---:|:------:|:----------:|
> | ResNet-50 (OTAM)       |  N/A    |     ✗     |  ✗  |  ✗  |  54.5  |    36.4    |
> | CLIP ViT-B (freeze)    |  N/A    |     ✗     |  ✗  |  ✗  |  58.2  |    29.5    |
> | BLIP ViT-B (freeze)    |  N/A    |     ✗     |  ✗  |  ✗  |  52.4  |    28.7    |
> | BLIP ViT-B (†)         |  N/A    |     ✓     |  ✗  |  ✓  |  53.7  |    32.2    |
> | BLIP ViT-B (‡)         |  N/A    |     ✓     |  ✓  |  ✓  | **58.4** | **33.3** |
> | BLIPv2 ViT-L (vanilla) |   ✓     |     ✗     |  ✗  |  ✗  |  67.0  |    40.5    |
> | BLIPv2 ViT-L (Ours)    |   ✓     |     ✓     |  ✓  |  ✓  | **82.3** | **54.1** |
>
> ---
>
> ### Key evidence from Table 1
>
> * **BLIP ViT-B (lighter backbone, closer to CLIP-FSAR).**
>   With BLIP ViT-B fixed, our modules increase performance from 52.4 / 28.7 (frozen) to 58.4 / 33.3 (Uni-Blend + LTQ + LSB), i.e., **+6.0** on HMDB51 and **+4.6** on SSv2-small. This shows that even on a lighter, more comparable backbone, the gains are predominantly methodological.
>
> * **BLIPv2 ViT-L (same 508M-parameter backbone).**
>   On BLIPv2 ViT-L, the vanilla Q-Former model (67.0 / 40.5) and our Uni-FSAR (82.3 / 54.1) share exactly the same backbone and parameter count; only UniQ-Former, Uni-Blend, LTQ, and LSB differ. The resulting **+15.3 / +13.6** improvements therefore isolate the contribution of our method rather than backbone scale.
>
> * **CLIP vs. BLIP baselines.**
>   The frozen CLIP ViT-B baseline (58.2 / 29.5) actually outperforms frozen BLIP ViT-B (52.4 / 28.7) on HMDB51 and SSv2-small, indicating that BLIP is not an inherently superior backbone in our configuration. The strong BLIP-based results thus cannot be attributed to BLIP being “simply stronger’’ than CLIP.

---

> > ### Author Response · Authors · 2025-11-20
> > **Response to Reviewer 1jCD - Part (2/3)**
> >
> > Comment: We thank R3 for raising this question about using static-image Q-Former queries for video. Below we clarify where temporal cues are actually modeled in Uni-FSAR and why static queries are sufficient for the FSAR setting we study.
> >
> > ---
> >
> > ### **[R3/W2] Static-image Q-Former queries for video actions**
> >
> > **Summary**
> >
> > * In Uni-FSAR, the 32 Q-Former visual queries are used **per frame** to extract spatial–semantic tokens, and temporal dynamics are modeled **afterwards** by a temporal transformer and OTAM-based alignment, not by Q-Former itself.
> > * The FSAR benchmarks we target (HMDB51, UCF101, Kinetics, SSv2-small) mainly involve **coarse, representative actions** where a few semantically key frames plus temporal ordering are sufficient. LTQ-guided Top-K selection is designed to focus supervision exactly on those key frames under frame-level ambiguity.
> > * Motion-aware VLM backbones (e.g., with explicit flow cues) would be a complementary enhancement, but are orthogonal to our contribution. Our aim here is to resolve frame-level ambiguity on top of a standard Q-Former backbone, which is supported by consistent gains over CLIP-FSAR and 3D/video baselines.
> >
> > ---
> >
> > ### **(A) Where temporal information is actually modeled in Uni-FSAR**
> >
> > In our design, Q-Former is deliberately used as a **per-frame spatial–semantic encoder**, and temporal reasoning is moved to later stages:
> >
> > * **Temporal transformer over Q-Former outputs.**
> >   For each video, we first obtain a sequence of frame-level tokens from Q-Former, one token per frame. This sequence is then processed by a multi-layer temporal transformer T to produce a sequence of t frame features V of dimension d. Self-attention in T computes pairwise relations between frames, so temporal patterns (onset, progression, transitions) are modeled at the **feature level** instead of raw pixels.
> >
> > * **OTAM for temporal alignment at inference.**
> >   OTAM aligns support and query sequences along monotonic paths, which is known to be robust to temporal shifts and local misalignment. Combined with the temporally processed frame features V, this adds another layer of temporal modeling on top of the static Q-Former features.
> >
> > In summary, Q-Former’s static queries answer **“what is in each frame”**, while **“how frames evolve”** is captured by the temporal transformer + OTAM stack guided by LTQ-based supervision.
> >
> > ---
> >
> > ### **(B) Why static queries are sufficient for the FSAR benchmarks we target**
> >
> > The reviewer’s concern is most critical when actions hinge on very fine-grained motion (e.g., subtle trajectories). In our FSAR setting, the benchmarks and our objective are slightly different:
> >
> > * **Coarse actions and short clips.**
> >   The datasets contain relatively short clips (around 3 seconds) with coarse action labels (e.g., *kick ball*, *smoke*, *laugh*, daily sports and interactions). For many classes, a small number of key frames already reveal the action via pose, interacted objects, and scene context.
> >
> > * **Main difficulty is frame-level ambiguity, not micro-motion.**
> >   As discussed in Sec. 1 and Fig. 1, the dominant challenge is redundant or misleading frames, rather than extremely subtle motion. Our uni-directional blending + LTQ + Top-K are designed specifically to **select** frames that best match the class semantics and **downweight** the rest.
> >
> > * **Empirical support.**
> >   Even with Q-Former used in this static way, Uni-FSAR consistently outperforms CLIP-FSAR and 3D/video baselines on HMDB51, UCF101, Kinetics, and SSv2-small (Tabs. 1–2), and shows strong cross-dataset transfer (Tab. 3). This indicates that, once coupled with our temporal modules, per-frame Q-Former features are sufficiently expressive for the level of temporal complexity present in these FSAR benchmarks.
> >
> > Thus, while Q-Former itself does not encode optical-flow-like motion, the combination of (i) temporally aware transformers, (ii) LTQ-guided frame selection, and (iii) OTAM alignment allows static queries to support effective video action understanding in our setting.
> >
> > ---
> >
> > ### **(C) Relation to motion-aware VLMs and future extensions**
> >
> > We regard motion-aware backbones (e.g., VLMs explicitly modeling optical flow or long-term trajectories) as **complementary** rather than conflicting:
> >
> > * Our contribution targets **frame-level ambiguity and prototype contamination**. The mechanisms we propose (uni-directional blending, LTQ, LSB, Top-K) are orthogonal and could, in principle, be applied to future motion-aware VLMs as well.
> > * Swapping the backbone to a motion-specialized VLM is therefore a natural extension, not a prerequisite for validating the current method. Under standard BLIP/BLIP-2 Q-Former backbones, the proposed architecture already yields substantial gains, indicating that static queries are an adequate basis for the specific problem we focus on.

---

> > > ### Author Response · Authors · 2025-11-20
> > > **Response to Reviewer 1jCD - Part (3/3)**
> > >
> > > Comment: We thank R3 for raising this point about the originality of our core components. Below we clarify what is inherited, what is new in the VLM-FSAR setting, and how the proposed combination forms a targeted solution to frame-level ambiguity rather than a loose collection of known tricks.
> > >
> > > ---
> > >
> > > ### **[R3/W3] On the novelty of uni-directional masking, LTQ, and Top-\$K\$ selection}**
> > >
> > > **Summary**
> > >
> > > * Our core components are the **uni-directional blending** mechanism, which explicitly targets frame-level ambiguity, and the LTQ-based **Semantic Bridging (LSB) loss**, which reduces the visual–textual semantic gap; **Top-K** selection and LTQ serve as supporting modules that make this pipeline effective in practice. **To our knowledge, this has not been explored in VLM-based FSAR**, where prior works typically rely on bi-directional blending with static text embeddings.
> > > * Top-K per se is not new, but our **LTQ-guided Top-K inside the LSB loss** is specifically designed to address the *frame-level ambiguity* problem we introduce in Sec. 1, and behaves differently from frame sampling alone.
> > > * Ablations and qualitative analyses show that this particular combination is essential to resolve prototype contamination and yields consistent gains over CLIP-FSAR and 3D/video baselines, suggesting that Uni-FSAR serves as a new baseline for ambiguity-aware FSAR rather than a straightforward reuse of existing designs.
> > >
> > > ---
> > >
> > > ### **(A) What is actually new in our FSAR setting**
> > >
> > > * **Uni-directional blending for frame-level ambiguity.**
> > >   Prior VLM-FSAR methods (e.g., CLIP-FSAR) use *bi-directional* blending, where text and frame tokens attend to each other. Uni-FSAR instead applies a **uni-directional mask** that lets queries attend to visual tokens but blocks attention into LTQ (Sec. 3.2). LTQ becomes a one-way “reader’’ of frames, explicitly designed to avoid prototype contamination under frame-level ambiguity.
> > >
> > > * **LTQ-based Semantic Bridging (LSB) loss.**
> > >   We introduce a learnable text query (LTQ) that aggregates frame evidence and, together with **LSB** (Sec. 3.2–3.3), uses LTQ–frame similarity to bridge visual features and class texts in a label-aware way. This directly targets the semantic gap behind ambiguous frames, beyond simply re-weighting static class embeddings used in prior FSAR/VLM work.
> > >
> > > * **LTQ-guided Top-K inside the loss.**
> > >   Top-K is not just a sampler but part of the objective: **LSB selects the Top-K frames per class by LTQ similarity and backpropagates only through them** (Sec. 3.3), implementing label-aware frame weighting for ambiguity. **Table 6** shows that LSB(Top-3) consistently outperforms GAP+Mean and Top-1 on HMDB51 and SSv2-small.
> > >
> > > ---
> > >
> > > ### **(B) Synergistic integration for frame-level ambiguity**
> > >
> > > Our goal is not merely to stack known components, but to produce a first FSAR baseline that *explicitly* tackles **frame-level ambiguity** (Sec. 1–2):
> > >
> > > * **Targeted to ambiguity.**
> > >   Uni-directional masking prevents irrelevant frames from feeding back into LTQ, LTQ encodes which frames are semantically aligned with the class, and LTQ-guided Top-K ensures that only those frames dominate the LSB supervision. This combination is designed to reduce **prototype contamination** from redundant or off-topic frames, which prior FSAR works do not explicitly handle.
> > >
> > > * **Empirical evidence.**
> > >   Ablations (**Table 6**) show that removing or weakening these components degrades performance, and attention/t-SNE visualizations (**Fig. 7; Appendix A.6**) illustrate that Uni-FSAR suppresses ambiguous frames where CLIP-FSAR still fires on background or overlapping actions. Overall we obtain an average **+2.3% Top-1 gain** across benchmarks, including **+6.5% on the ambiguity-prone HMDB51** dataset.

---

### Official Review · Reviewer_kEE2 · 2025-10-30

**Soundness:** 2
**Presentation:** 2
**Contribution:** 2
**Rating:** 4
**Confidence:** 3

**Summary:**

The Uni-FSAR framework introduces a Uni-Directional Blending (UDB) mechanism such that visual queries (VQs) can attend to a learnable text query (LTQ), but not vice versa. Here Top-K most relevant frames are aligned with the LTQ to suppress noisy or irrelevant frames and combined with a Learnable Text Query-based Semantic Bridging (LSB) loss,
The model is further integrated into an OTAM-based pipeline for few-shot action recognition. While the proposed Uni-FSAR framework presents an interesting architecture for few-shot video understanding, there are some conceptual and methodological weaknesses which limit its claim to effectively handle frame-level ambiguity.

**Strengths:**

1. The use of a learnable text query avoids reliance on handcrafted textual prompts.
2. The LSB loss encourages semantic selectivity and yields better interpretability through attention visualization.

**Weaknesses:**

1. Even after Top-K selection in the loss function, all frame embeddings are averaged equally to form the video prototype. This contradicts the goal of mitigating frame-level redundancy — the prototype is still influenced by uninformative frames.
2. The choice of Top-K  and the loss weighting factor α  seems to be dataset-specific and tuned manually. This undermines claims of generalization.
3. Despite claiming to address frame-level ambiguity, the paper introduces no mechanism  that measures inter-frame differences (e.g., variance). The Top-K selection in the LSB loss merely filters frames based on similarity to the LTQ, but it does not take into account the embedding variance between frames within a video.

**Questions:**

I am wondering  how without ever measuring or minimizing frame-level differences in the learning objective it is possible to  “handle frame-level ambiguities”, maybe  I am missing something,  then kindly clarify!

---

> ### Author Response · Authors · 2025-11-20
> **Response to Reviewer kEE2 - Part (1/4)**
>
> **Comment:**
> We thank R2 for raising the question of how Uni-FSAR handles frame-level ambiguity without an explicit variance term in the loss. Below we clarify (i) how inter-frame differences are implicitly modeled in the feature space, and (ii) how our new variance analysis in below Table quantitatively supports this behavior.
>
> ---
>
> ### **[R2/Q1 & W3] Handling frame-level ambiguity without an explicit variance term**
>
> **Summary**
>
> * Uni-FSAR does not add a hand-crafted pairwise variance term to the loss. Instead, frame-level ambiguity is handled **in the learned feature space**: a temporal transformer encodes inter-frame relations, and the LTQ-guided Top-$K$ + LSB loss acts as a *semantic, label-aware frame weighting* mechanism that amplifies informative frames and down-weights redundant or off-topic ones.
> * To directly address the concern about “measuring” frame-level differences, we report **feature-space variance statistics** on HMDB51 with a fixed backbone below Table. After training Uni-FSAR, feature variance across frames and mean pairwise feature distance increase by **+302.6%** and **+123.0%**, respectively, while temporal consistency between adjacent frames decreases only slightly (from 0.9567 to 0.9077), remaining high. This shows that the model learns to separate frames more discriminatively **without** destroying temporal coherence.
> * An explicit low-level variance regularizer could also be a reasonable design choice, but in our setting it may become highly dataset- and domain-dependent (e.g., sensitive to frame rate, trimming quality, or camera motion). Our LTQ-based Top-$K$ + LSB strategy instead offers a complementary, more **semantic** way to modulate inter-frame dispersion, and the same training recipe still transfers well to datasets with fewer redundant frames such as SSv2-small.
>
> ---
>
> ### **(A) How Uni-FSAR implicitly models inter-frame differences**
>
> In Uni-FSAR, frame-level ambiguity is handled in the **feature space** rather than via an explicit variance penalty:
>
> * **Temporal transformer.**
>   As described in Sec. 3.2, the temporal transformer $T$ takes the sequence of frame features (one feature vector per frame) and updates each frame via self-attention. This directly models pairwise similarities and differences between frames, so inter-frame relations are encoded in the backbone representation $V \in \mathbb{R}^{t \times d}$.
>
> * **LTQ-guided Top-$K$ + LSB as semantic weighting.**
>   In Sec. 3.3, the LTQ-based Semantic Bridging (LSB) loss computes similarities between frame embeddings $V$ and class text embeddings $F_\psi$, and only the Top-$K$ frames per class contribute to $L_{\text{LSB}}$. These selected frames receive strong gradients toward the class prototype, while ambiguous or redundant frames receive almost no update, inducing a **label-aware dispersion** rather than uniformly shrinking variance across all frames.
>
> This implicitly “measures and uses” inter-frame differences at the representation level and matches our t-SNE and qualitative results, where Uni-FSAR forms more discriminative class clusters and suppresses ambiguous frames compared to CLIP-FSAR.
>
> ---
>
> ### **(B) Quantitative feature-space variance analysis**
>
> To make the discussion more concrete, we explicitly measure variance-related statistics on frame embeddings **before and after** training Uni-FSAR on HMDB51, using the *same* videos and *same* backbone:
>
> | **Metric** | **Before training** | **After training** | **Relative change** |
> | --- | --- | --- | --- |
> | Feature variance (across frames) | 0.0099 | 0.0400 | +302.6% |
> | Mean pairwise feature distance | 0.0607 | 0.1354 | +123.0% |
> | Temporal consistency (adjacent cos.) | 0.9567 | 0.9077 | -5.1% |
>
> These feature-space changes accompany the Top-1 accuracy gains on HMDB51 reported in Table~1, where frame-level ambiguity is particularly severe, and the same training strategy also improves performance on SSv2-small, where videos are temporally well trimmed. Together, this suggests that our implicit handling of inter-frame differences is **beneficial and not overfitted** to a specific dataset.
>
> **We kindly note that an additional visualization of this quantitative result is provided in the newly uploaded supplementary file.**

---

> ### Author Response · Authors · 2025-11-20
> **Response to Reviewer kEE2 - Part (2/4)**
>
> ### **(C) Why no explicit variance regularizer in this work**
>
> While one could in principle add an explicit frame-difference or variance term to the objective, such terms tend to be tightly coupled to video style, frame rate, camera motion, and trimming quality. In this work, we therefore favor a **semantic, LTQ-based** mechanism that:
>
> * adjusts inter-frame dispersion according to alignment with class text and temporal context, rather than low-level thresholds, and
> * remains effective across datasets with different levels of redundancy (HMDB51, UCF101, Kinetics, SSv2-small).
>
> Designing an explicit, domain-agnostic variance-aware regularizer that complements our LTQ-guided weighting is an interesting direction for future work. Our current analysis shows that even without such a term, the model **learns and exploits inter-frame differences in feature space** to handle frame-level ambiguity.

---

> ### Author Response · Authors · 2025-11-20
> **Response to Reviewer kEE2 - Part (3/4)**
>
> Comment: We thank R2 for pointing out the Top-$K$ frame selection in the loss and uniform averaging in the prototype. Below we clarify how prototypes are implicitly biased toward informative frames, and why an explicit Top-$K$-only prototype actually hurts performance.
>
> ### **[R2/W1] Prototype averaging vs. frame-level redundancy**
>
> **Summary**
>
> - Although the final prototype is computed by averaging all frame embeddings, LTQ-guided Top-$K$ + LSB is applied *only* to the most label-consistent frames during training. This gives strong gradients to informative frames and almost none to redundant ones, so informative frames dominate the learned prototype direction even under uniform averaging.
> - Ablation without Top-$K$ (all frames participate in LSB) degrades performance by about $3.2$\% on HMDB51 5-shot (Table~6 in the paper), showing that selective gradients are necessary for mitigating redundancy.
> - A new “Top-$K$-only prototype” variant, where the prototype itself is computed from Top-$K$ frames instead of all frames, performs **12-17 points worse** across datasets and shots (table below), indicating that hard pruning at prototype time overfits and loses useful context compared to our indirect, training-time weighting strategy.
>
> ---
>
> ### **How selective gradients bias prototypes despite uniform averaging**
>
> In Uni-FSAR, the LTQ-based Semantic Bridging (LSB) loss is applied only to Top-$K$ frames that are most consistent with the class text:
>
> - For each class, we compute similarities between frame embeddings $V$ and the class text embedding and select the Top-$K$ frames (Sec. 3.3).
> - Only these Top-$K$ frames contribute to $L_{\text{LSB}}$; non-selected frames receive essentially **zero gradient** from this term.
>
> Over training, this induces *unequal* updates across frames:
>
> - semantically informative frames are repeatedly pulled toward the class prototype in the embedding space,
> - ambiguous or redundant frames stay farther from the class text and are weakly updated.
>
> Thus, when we later average all frame embeddings to form the prototype, the resulting direction is already biased toward the informative frames shaped by LSB, while redundant frames contribute much less in a discriminative sense.
>
> ---
>
> ### **Evidence from ablations and Top-$K$-only prototype variant**
>
> *Top-$K$-only prototype variant.* To directly address the concern about averaging, we implemented a variant where the prototype itself is computed only from Top-$K$ frames (instead of averaging all frames) while keeping the rest of the pipeline unchanged. Results averaged over 10k episodes are:
>
> | Dataset | Shot | Original Acc (%) | Top-$K$-only Acc (%) | $\Delta$ (%) |
> | --- | --- | --- | --- | --- |
> | HMDB51 | 1 | 82.3 | 65.5 | -16.8 |
> | HMDB51 | 5 | 90.5 | 76.8 | -13.7 |
> | SSv2-small | 1 | 54.1 | 41.4 | -12.7 |
> | SSv2-small | 5 | 65.2 | 52.3 | -12.9 |
>
> **Top-$K$-only prototype vs. original prototype (all frames) under the same training setup.**
>
>
> Across both HMDB51 (high frame-level ambiguity) and SSv2-small (clean ambiguity), using only Top-$K$ frames at prototype time consistently hurts performance by 12-17 points. This indicates that:
>
> - hard discarding non–Top-$K$ frames at inference loses useful contextual information and leads to overfitting to a small subset of frames, especially in low-shot regimes;
> - our design choice,*selective gradients during training + full averaging at inference*, strikes a better balance: it suppresses the influence of noisy frames in the learned representation while still leveraging broader temporal context when forming prototypes.
>
> Taken together, these results show that the mitigation of frame-level redundancy in Uni-FSAR is achieved *indirectly* through LTQ-guided Top-$K$ training, rather than through a hard Top-$K$ prototype, and that this indirect strategy is empirically more effective.

---

> ### Author Response · Authors · 2025-11-20
> **Response to Reviewer kEE2 - Part (4/4)**
>
> **Comment:**
> We thank R2 for raising the concern that Top-$K$ and the loss weight $\alpha$ might be dataset-specific knobs rather than stable design choices. Below we clarify how $K$ is fixed globally, how $\alpha$ shows low sensitivity, and why our generalization claims do not rely on narrow hyperparameter tuning.
>
> ---
>
> ### **[R2/W2] Top-$K$ and $\alpha$: dataset-specific tuning concern**
>
> **Summary**
>
> * The Top-$K$ value is **not** tuned per dataset: we set $K = 3$ once based on an ablation in Table~6 and then use the same value across all benchmarks and cross-dataset evaluations.
> * The loss weight $\alpha$ is an $\mathcal{O}(1)$ balance term (following CLIP-FSAR), and our additional sensitivity study with $\alpha \in \{1.0, 1.5, 3.0\}$ shows only about $0.2$–$0.5$ Top-1 variation, indicating low sensitivity.
> * The strong gains on multiple datasets and cross-dataset transfers (Table~3) are therefore attributed to the Uni-FSAR design (uni-directional blending, LTQ, LSB), not to finely tuned, dataset-specific $(K,\alpha)$ choices.
>
> ---
>
> ### **(A) Top-$K$ is fixed globally, not re-tuned per dataset**
>
> * In the main paper (Table~6), we compare GAP+Mean, LSB(Top-1), and LSB(Top-3), and LSB(Top-3) consistently achieves the best performance on both HMDB51 and SSv2-small.
> * Based on this single ablation and the intuition that a few key frames dominate the semantics of a short video, we **fix** $K = 3$ once and use it for all experiments (HMDB51, UCF101, Kinetics, SSv2-small, and cross-dataset evaluation).
> * We do not re-tune $K$ for different datasets, shot numbers, or transfer settings, so Top-$K$ acts as a **global design choice** rather than a dataset-specific hyperparameter.
>
> ---
>
> ### **(B) $\alpha$ is a stable balance term with low sensitivity**
>
> In Eq. (15), the overall loss is written as
> $L = L_{\mathrm{LSB}} + \alpha \, L_{\mathrm{proto}}$,
> where $L_{\mathrm{LSB}}$ is the LTQ-based Semantic Bridging loss and $L_{\mathrm{proto}}$ is the prototype-alignment loss. Following CLIP-FSAR, $\alpha$ is chosen to roughly balance the magnitudes of the two terms per dataset (Appendix A.2, Table 9) and kept in a small $\mathcal{O}(1)$ range without exhaustive tuning.
>
> To directly examine sensitivity, we run an additional study on HMDB51 and SSv2-small (5-way 1-shot) with $\alpha \in \{1.0, 1.5, 3.0\}$ under the full Uni-FSAR setting (BLIPv2 ViT-L, UniQ-Former, LTQ, LSB, Top-3):
>
> | **$\alpha$** | **HMDB51 1-shot** | **SSv2-small 1-shot** |
> |--------------|-------------------|------------------------|
> | 1.0          | 82.1              | 53.3                   |
> | 1.5          | 82.0              | 53.8                   |
> | 3.0          | 81.9              | 53.6                   |
> | **Δ**        | **±0.2**          | **±0.5**               |
>
> On HMDB51, accuracy changes only by about 0.2 points, and on SSv2-small within 0.5 points. This indicates that Uni-FSAR is **robust** to reasonable changes of $\alpha$ and that our gains do not rely on finely tuned, dataset-specific weighting.
>
> ---
>
> ### **(C) Generalization evidence beyond a specific $(K,\alpha)$ pair**
>
> Finally, our “generalizable under ambiguity’’ claim is supported not only by within-dataset benchmarks (Tables 1,2) but also by cross-dataset transfer (Table 3). For example, when transferring from HMDB51$\rightarrow$SSv2-small and SSv2-small$\rightarrow$HMDB51, Uni-FSAR significantly outperforms CLIP-FSAR in both 1-shot and 5-shot settings (e.g., +18.4 pts in HMDB51$\rightarrow$SSv2-small 1-shot), using the same fixed configuration ($K = 3$ and the corresponding $\alpha$ from Appendix~A.2) without any re-tuning on the target dataset.
>
> Taken together, the globally fixed $K = 3$, the low sensitivity to $\alpha$, and the strong cross-dataset gains indicate that the robustness of Uni-FSAR comes from its architectural design (uni-directional blending, LTQ, LSB), rather than from dataset-specific hyperparameter tuning.

---

### Official Review · Reviewer_MdP9 · 2025-11-01

**Soundness:** 2
**Presentation:** 3
**Contribution:** 3
**Rating:** 6
**Confidence:** 3

**Summary:**

This paper proposes a Uni-FSAR framework that aims to improve prototype construction under frame-level ambiguity by selectively utilizing semantically relevant frame information.
The authors design a uni-directional blending strategy to prevent irrelevant frames from contaminating class prototypes and introduce a Learnable Text Query (LTQ) module to achieve semantic alignment between visual features and class labels.

**Strengths:**

1.The motivation is clear .
2.The paper is very clearly written. The methodology is presented in a logical and accessible manner, with well-organized sections, clear mathematical formulations, and informative visualizations that make the technical design easy to follow.

**Weaknesses:**

1.The main comparison is made against CLIP-FSAR, yet all experiments in this paper adopt a more powerful BLIP backbone, making it difficult to disentangle whether the performance gain stems from the stronger backbone or from the proposed method itself.

2.The paper lacks a clear ablation study on the Uni-directional Blending mechanism.
A comparison among uni-directional, bi-directional, and reverse-directional attention designs would be necessary to verify that the observed improvements truly correspond to the claimed motivation.

**Questions:**

The motivation of this work is meaningful, and the results indeed demonstrate improved performance.
However, it remains unclear why the uni-directional blending strategy can selectively integrate relevant frames while preventing prototype contamination by irrelevant visual noise.
Please clarify how the proposed mechanism theoretically or empirically achieves this selective filtering, and explicitly connect the motivation to the method’s operational design.

---

> ### Author Response · Authors · 2025-11-20
> **Response to Reviewer MdP9 - Part (1/3)**
>
> Comment: We thank R1 for the thoughtful question on how our uni-directional blending strategy can both select relevant frames and avoid prototype contamination. We clarify the mechanism and its empirical support below.
>
> ---
>
> ### [R1/Q1] Uni-directional blending and selective frame usage
>
> **Summary**
>
> - Uni-directional blending makes the learnable text query (LTQ) a *one-way semantic reader*, so class text does not overwrite or blur the differences between frame features.
> - The Top-K LTQ-based Semantic Bridging (LSB) loss sends strong gradients only to frames that are highly consistent with the class text, while noisy/background frames receive almost no supervision from this term.
> - As a result, even simple averaging yields prototypes dominated by action-relevant frames rather than visual noise, which matches the observed gains, especially on ambiguity-heavy actions.
>
> ---
>
> ### (1) Uni-directional blending as a one-way semantic reader
>
> **Motivation.**
> We want the class text to *read* which frames are relevant, without broadcasting its information back into every frame. If text freely flows into frame tokens, frame features become homogenized and prototypes are easily contaminated by irrelevant frames.
>
> **Mechanism.**
> LTQ and frame-wise visual queries are processed together, but the attention mask is defined so that all queries can attend to visual features while no query can attend to LTQ itself. LTQ aggregates evidence from frames, yet never becomes a source that modifies them.
>
> **Effect.**
> When we enable only this uni-directional blending with LTQ (without LSB), HMDB51 1-shot accuracy increases from 67.0\% to 80.2\%, with similar trends on SSv2-Small. This shows that preventing text-to-frame leakage already reduces prototype contamination and improves how relevant frames are encoded.
>
> ---
>
> ### (2) Top-K LSB focusing gradients on label-consistent frames
>
> **Motivation.**
> In few-shot action videos, only a few frames truly express the labeled action, while many are neutral or ambiguous. We need a mechanism that concentrates supervision on those informative frames instead of treating all frames equally.
>
> **Mechanism.**
> After temporal modeling, we compute the similarity between each frame feature and the class text and keep only the Top-K frames per video (K = 3) when computing LSB. Gradients from LSB are applied only to these frames; non-selected frames receive zero gradient from this term.
>
> **Effect.**
>  This acts as an explicit frame-level filter during optimization: informative frames are repeatedly pulled closer to the class text in the joint space, while irrelevant frames remain weakly aligned. In ablations, replacing plain global averaging with Top-K LSB improves accuracy across datasets and also yields consistent gains on SSv2-Small, indicating sharper, more discriminative representations.
>
> ---
>
> ### (3) Prototypes biased toward relevant frames, observed in practice.
>
> **Mechanism.**
> Class prototypes are obtained by averaging the temporally modeled frame features and then matching them to query videos. Because (i) uni-directional blending avoids text flooding frame features, and (ii) Top-K LSB strongly aligns the truly informative frames, these frames dominate the average.
>
> **Effect.**
> The resulting prototypes are biased toward actual action segments rather than background or incidental content. The full model (uni-directional blending + LTQ + LSB) achieves the best performance across datasets, with the largest gains on ambiguity-prone classes such as “kick ball” and “smoke”. Qualitative results further show that Uni-FSAR activates mainly on true action frames, while the baseline often fires on faces or static background, matching the selective behavior our design aims to achieve.

---

> > ### Author Response · Authors · 2025-11-20
> > **Response to Reviewer MdP9 - Part (2/3)**
> >
> > Comment: We thank R1 for raising this important fairness concern regarding backbone strength versus methodological contributions. Below we explain how Table 1 disentangles these factors and why its key numbers indicate that the gains mainly come from our method, not from backbone scaling.
> >
> > ---
> >
> > ### **[R1/W1] Backbone fairness: disentangling method gains from backbone capacity**
> > To make this explicit, below Table 1  fixes the backbone within each BLIP variants and incrementally adds our components (1-shot setting).
> >
> > | Backbone                 | Q-Former | Uni-Blend | LTQ | LSB | HMDB51 | SSv2-Small |
> > |--------------------------|:--------:|:---------:|:---:|:---:|:------:|:----------:|
> > | ResNet-50 (OTAM)         |   N/A    |    ✗      | ✗   | ✗   |  54.5  |    36.4    |
> > | CLIP ViT-B (freeze)      |   N/A    |    ✗      | ✗   | ✗   |  58.2  |    29.5    |
> > | BLIP ViT-B (freeze)      |   N/A    |    ✗      | ✗   | ✗   |  52.4  |    28.7    |
> > | BLIP ViT-B (†)           |   N/A    |    ✓      | ✗   | ✓   |  53.7  |    32.2    |
> > | BLIP ViT-B (‡)           |   N/A    |    ✓      | ✓   | ✓   | **58.4** | **33.3** |
> > | BLIPv2 ViT-L (vanilla)   |    ✓     |    ✗      | ✗   | ✗   |  67.0  |    40.5    |
> > | BLIPv2 ViT-L (Ours)      |    ✓     |    ✓      | ✓   | ✓   | **82.3** | **54.1** |
> >
> > ---
> > **Summary**
> >
> > * On the lighter **BLIP ViT-B** backbone (closer in capacity to CLIP-FSAR), fixing the backbone and adding our modules yields clear 1-shot gains from **52.4 / 28.7** (frozen BLIP ViT-B) to **58.4 / 33.3** (Uni-Blend + LTQ + LSB), i.e., **+6.0%** on HMDB51 and **+4.6%** on SSv2-Small *within the same backbone*.
> > * On the stronger **BLIPv2 ViT-L** backbone, again with the backbone fixed, our full model improves from **67.0 / 40.5** (vanilla Q-Former) to **82.3 / 54.1**, corresponding to **+15.3%** and **+13.6%** 1-shot gains purely from UniQ-Former, Uni-Blend, LTQ, and LSB.
> > * These intra-backbone improvements, especially the large margins on BLIPv2 ViT-L, are much larger than typical backbone-only swaps and cannot be attributed to backbone capacity alone, indicating that the proposed method is the dominant source of the reported performance gains.

---

> > > ### Author Response · Authors · 2025-11-20
> > > **Response to Reviewer MdP9 - Part (3/3)**
> > >
> > > Comment: We thank R1 for pointing out the need for an explicit ablation on the Uni-directional Blending mechanism. We report a controlled comparison where only the attention direction is varied and show that the key difference, especially under frame-level ambiguity, appears in the attention patterns rather than in large numeric gaps.
> > >
> > > ---
> > >
> > > ### **[R1/W2] Ablation on uni-/bi-/reverse-directional blending**
> > >
> > > **Summary**
> > >
> > > * We ablate three variants within the same Uni-FSAR pipeline on **BLIPv2 ViT-L**, changing **only** the attention direction (uni-directional, bi-directional, reverse-directional) while keeping LTQ, LSB, and Top-K selection identical. This isolates the effect of the blending mechanism itself.
> > > * Uni-directional blending achieves the best Top-1 accuracy of **82.3%**, while the bi-directional variant is consistently lower at **81.2%**, and the reverse-directional variant collapses to around **20.0%**. Thus, allowing text-to-frame feedback does not improve performance and can destabilize training as our initial claim.
> > > * More importantly, frame–class attention heatmaps on our frame-level ambiguity cases (**please see the additional visualization figure in the supplementary material**) show that bi-/reverse-directional variants spread or misplace attention over non-target classes, whereas our uni-directional design sharply focuses on the correct class and a few informative frames. This qualitatively explains why uni-directional blending better handles ambiguous frames despite the modest numeric gap.
> > >
> > > ---
> > >
> > > ### **Quantitative ablation on attention direction**
> > >
> > > We implement three attention designs under the same backbone (BLIPv2 ViT-L) and training protocol. All variants share the same LTQ representation, LSB loss, and Top-K frame selection; only the attention direction is altered:
> > >
> > > | **Attention design**                          | **Top-1 Acc. (%)** |
> > > |----------------------------------------------|---------------------|
> > > | Bi-directional (CLIP-FSAR style)             | 81.2                |
> > > | Reverse-directional (text → frames)          | 20.0                |
> > > | Uni-directional (ours, frames → text)        | **82.3**            |
> > >
> > > Although the gap between uni- and bi-directional variants is numerically modest (82.3% vs. 81.2%), it is consistent across runs and appears together with a clear difference in attention behaviour, as discussed next.
> > >
> > > ---
> > >
> > >
> > > ### **Qualitative evidence on frame-level ambiguity**
> > >
> > > The **new supplementary material** includes frame–class attention visualizations for the same HMDB51 episode (8 frames, all classes). In the bi-directional and reverse-directional variants, attention remains diffuse and often highlights non-target classes (e.g., *kiss*, *sword*, *flic-flac*, *chew*, *pour*, *smoke*, *wave*), while the true class (e.g., *laugh*) does not form a clear, dominant stripe. In contrast, our uni-directional blending concentrates attention on the correct class and a small subset of informative frames, strongly suppressing other classes.
> > >
> > > This pattern is consistently observed across additional qualitative examples in **Appendix A.6** and shows that bi-directional designs are more vulnerable to our frame-level ambiguity cases, whereas uni-directional blending better isolates truly informative frame–class interactions. Thus, the combination of (i) higher and more stable Top-1 accuracy and (ii) sharply focused attention maps supports our motivation for using uni-directional blending.

---

### Author Response · Authors · 2025-11-20
**General response to all reviewers**

### **First, we would like to thank all reviewers for their thoughtful and constructive feedback.**

**Your comments motivated us to validate our intuition with additional experiments and more careful theoretical analysis, and we genuinely enjoyed engaging with your questions during the rebuttal phase.**

We also provide further qualitative visualizations in the **newly uploaded supplementary PDF** and we kindly encourage the reviewers to refer to these materials for a more intuitive understanding of our method.

---

> ### Author Response · Authors · 2025-11-28
> **Open Invitation to Further Discussion**
>
> ### **We truly appreciate the reviewers’ time and thoughtful engagement with our submission.**
>
> Your feedback has already helped us clarify and improve several aspects of the work.
>
> Please feel free to raise **any additional questions or concerns at any point**,  we welcome all forms of discussion, whether major or minor.
>
> **We will continue to respond promptly and do our best to provide clear evidence and explanations throughout the discussion period.**

---

### Author Response · Authors · 2025-11-30
**Author Rebuttal Summary for AC (part 1/2)**

**To assist the Area Chair**, we briefly summarize below how our rebuttal addresses the five main concerns regarding: **(1) Backbone Fairness, (2) Ambiguity Handling Mechanism, (3)  Necessity of LTQ, LSB, and Top-K , (4) Hyperparameter Robustness, and (5) Temporal Information.**

In preparing these responses, we sincerely thank all reviewers for their thoughtful feedback. We are particularly encouraged by the recognition of our **well-grounded motivation and sharp problem formulation** (R1, R3, R4), **the semantic selectivity and interpretability enabled by LTQ + LSB** (R2), and **robust performance** across noisy and cross-dataset benchmarks (R3, R4).

We have uploaded a **Revised Paper** and **Supplementary Material** (including new attention visualizations) reflecting these updates. Detailed responses to the common concerns are provided below.


---

### **1. Backbone Fairness: Disentangling Method from Capacity**
*(Addressing R1 MdP9, R3 1jCD)*

Reviewers expressed concern that the performance gap between Uni-FSAR (BLIPv2 ViT-L) and CLIP-FSAR (ViT-B) might stem solely from the larger backbone. To address this, we performed a controlled analysis **fixing the backbone** (Table 1 below):

**Table 1: Controlled Fairness Analysis (1-shot).** We fix the backbone and incrementally add our modules.
| Backbone | Method | HMDB51 (1-shot) | SSv2-small (1-shot) |
| :--- | :--- | :---: | :---: |
| **CLIP ViT-B (Frozen)** | Baseline (CLIP-FSAR) | 58.2 | 29.5 |
| **BLIP ViT-B (Frozen)** | Baseline | 52.4 | 28.7 |
| **BLIP ViT-B (Frozen)** | **+ Ours (Uni-Blend, LTQ, LSB)** | **58.4 (+6.0)** | **33.3 (+4.6)** |
| | | | |
| **BLIPv2 ViT-L** | Baseline (Vanilla Q-Former) | 67.0 | 40.5 |
| **BLIPv2 ViT-L** | **+ Ours (Full Uni-FSAR)** | **82.3 (+15.3)** | **54.1 (+13.6)** |

---

* **Lighter Backbone (BLIP ViT-B):** On a backbone comparable to CLIP-FSAR, enabling our modules yields gains of **+6.0%** and **+4.6%**, proving methodological effectiveness.
* **Larger Backbone (BLIPv2 ViT-L):** Comparing the vanilla Q-Former baseline to Uni-FSAR (both using ViT-L) shows a massive gain of **+15.3%**, purely from our method.
* **Frozen Baseline Check:** The frozen CLIP ViT-B (58.2%) outperforms frozen BLIP ViT-B (52.4%), proving that BLIP is **not** inherently superior.

**Conclusion:** The improvements are driven by the Uni-FSAR design, not backbone scaling. We kindly remind you that this controlled analysis is **already provided** in Table 13 of Appendix A.5."

---

### **2. How Uni-FSAR Handles Frame-Level Ambiguity**
*(Addressing R1 MdP9, R2 kEE2, R4 6P94)*

We clarify how our mechanism selectively filters noise without explicit variance terms:

**Uni-directional Blending:** Acts as a **"one-way semantic reader."** As requested, we performed additional ablation studies showing that Uni-directional attention reaches **82.3%**, while **Reverse-directional** collapses to ~20%. **Additional qualitative heatmaps (at Supplementary)** confirm that our design focuses sharply on action-relevant frames, whereas **bi-directional** attention diffuses over background noise.

**Implicit Variance Modeling**: Following the reviewer's suggestion to verify ambiguity handling, we measured feature-space statistics before and after training. Uni-FSAR increases feature variance by +302.6% and **pairwise distance by +123.0%**, confirming that the model learns to separate frames discriminatively without a handcrafted penalty.

---

### **3. Necessity of LTQ, LSB, and Top-K**
*(Addressing R3 1jCD, R4 6P94)*

Our components are not generic variations but are specifically tailored for FSAR ambiguity:
* **LTQ (Learnable Text Query):** Unlike static prompts in prior works, LTQ is updated by video frames to bridge the **granularity gap** (phrase vs. frame) and **domain gap** (web image vs. noisy clip).
* **LTQ-guided Top-K:** Unlike the text-agnostic samplers in CNN-based FSAR (e.g., [1]), our Top-K acts as a **text-conditioned gradient gate** inside the LSB loss, supervising only label-consistent frames.

[1] Task-adaptive Spatial-Temporal Video Sampler for Few-shot Action Recognition

---

### **4. Robustness: Top-K and Hyperparameters**
*(Addressing R2 kEE2, R4 6P94)*

* **Global Top-K:** We fix $K=3$ globally based on ablations. We **do not** re-tune $K$ per dataset.
* **Sensitivity:** Varying the loss weight $\alpha$ results in negligible fluctuations ($\pm 0.2-0.5\%$), confirming robustness.
* **"Top-K Only Prototype" Failure:** To directly address the concern regarding prototype averaging, we tested a variant that uses *only* Top-K frames for the prototype (discarding others). This **degraded performance by 12–17 points**, validating our design choice: *Selective Gradients (Training) + Full Averaging (Inference).*

---

---

> ### Author Response · Authors · 2025-11-30
> **Author Rebuttal Summary for AC (part 2/2)**
>
> ### **5. Preservation of Temporal Information & Generalization**
> *(Addressing R3 1jCD, R4 6P94)*
>
> * **Motion is Preserved:** Top-K applies only to the auxiliary LSB loss. **All frames** (selected or not) pass through the Temporal Transformer and OTAM. **No temporal cues are discarded** (as pointed out by R4) from the inference pipeline.
> * **Evidence from SSv2-small:** On this fine-grained dataset, Uni-FSAR matches baselines in 1-shot and outperforms them in 5-shot (+7.0%), proving it handles subtle motion well.
> * **"Generalizable under Ambiguity":** This claim is supported by large cross-dataset gains (e.g., **+18.4%** on HMDB51 $\to$ SSv2), showing robust transfer between ambiguity-heavy and ambiguity-light domains.
>
> ---
>
> We hope this summary resolves the major concerns. Detailed responses to specific questions are provided in the individual threads below.
>
> Best regards,
> **The Authors**

---

### Meta-Review · Area_Chair_tRN7 · 2026-01-08

**Summary:**

Main concerns raised include
1) The use of a more powerful BLIP backbone may raise fairness issue in comparison and examine the exact effect of proposed method

2) Lacking ablation study of key components like the Uni-directional Blending mechanism.

3) The rationale behind: Why can the proposed uni-directional blending strategy selectively integrate relevant frames while preventing prototype contamination by irrelevant visual noise?

4) Evidence on frame-level ambiguity or frame selection

5) Top-K selection in the loss function vs frame selection: how they are made consistent?

6) Limited novelty, as all core components (uni-directional attention masking, learnable text queries, and Top-K frame selection) are well-established techniques from existing work

**Reviewer Concerns:**

To which degree each of these issues has been addressed:

1) This issue is still out there, not addressed in the revised version. For main tables (Tables 1-2) in paper, the comparison should be made under the same backbone used, as the backbone would impose a big impact on the performance. The Backbone fairness table can only verify the generality of the proposed design but not address the fairness of comparing with previous alternative models. About searching label-consistent frames, it is unclear how to more effectively informative frames.

Also, the compared methods are not up to date, with quite a number of more recent works excluded, for example, the following works which can be quickly found in google scholar.

*Several recent works that should be compared*:
* Frame Order Matters: A Temporal Sequence-Aware Model for Few-Shot Action Recognition. AAAI 2025
* Beyond Label Semantics: Language-Guided Action Anatomy for Few-shot Action Recognition. ICCV 2025
* Trokens: Semantic-Aware Relational Trajectory Tokens for Few-Shot Action Recognition. ICCV 2025

2) The authors have done the experiments as suggested (which is necessary for this work as this design is claimed as one key design), but the analysis is limited, for example, why does the variant of text -> frame collapse?

3) Still, the response is vague about how and why the uni-directional blending can prevent contamination of prototype, just staying at the level of expressing the intention. This motivation, "We want the class text to read which frames are relevant", is too general to claim novelty.

4) Added visualisation and examples but the analysis is shallow and less convincing. As expected, for each example, the authors should discuss why one frame is informative or not, if it is picked up or not, and how many irrelevant frames are filtered out successfully, connecting to the proposed design.

5) The authors explained further, but the connection from LTQ-guided Top-K + LSB loss to label-aware frame weighting via a temporal transformer is rather implicit and hard to understand/capture. This interpretation is hardly convincing and transparent. Using the feature-space variance statistics as the metric of frame ambiguity is not intuitive. Overall, the whole presentation is ambiguous and confusing conceptually.

6) The authors argued but that cannot change the fact of those being existing techniques, and this argument is generally not strong.

**Reviewer Scores:**

While the authors have made great efforts to response, as it stands, this work is limited in a number of aspects, including the interpretation of the proposed model, novelty, comparison fairness, and absence of more recent alternatives. With these issues, it is hard to convince the reviewers  as well as Acs.

This work is clearly below what is expected.

---

### Decision · Program_Chairs · 2026-01-26

Reject